# Molecular mechanism for vitamin C-derived C$^5$-glyceryl-methylcytosine DNA modification catalyzed by algal TET homologue CMD1

Wenjing Li[1,4], Tianlong Zhang[1,4], Mingliang Sun[1,4], Yu Shi[1,2], Xiao-Jie Zhang[1], Guo-Liang Xu[1] & Jianping Ding [1,2,3✉]

C$^5$-glyceryl-methylcytosine (5gmC) is a novel DNA modification catalyzed by algal TET homologue CMD1 using vitamin C (VC) as co-substrate. Here, we report the structures of CMD1 in apo form and in complexes with VC or/and dsDNA. CMD1 exhibits comparable binding affinities for DNAs of different lengths, structures, and 5mC levels, and displays a moderate substrate preference for 5mCpG-containing DNA. CMD1 adopts the typical DSBH fold of Fe$^{2+}$/2-OG-dependent dioxygenases. The lactone form of VC binds to the active site and mono-coordinates the Fe$^{2+}$ in a manner different from 2-OG. The dsDNA binds to a positively charged cleft of CMD1 and the 5mC/C is inserted into the active site and recognized by CMD1 in a similar manner as the TET proteins. The functions of key residues are validated by mutagenesis and activity assay. Our structural and biochemical data together reveal the molecular mechanism for the VC-derived 5gmC DNA modification by CMD1.

[1] State Key Laboratory of Molecular Biology, Shanghai Institute of Biochemistry and Cell Biology, Center for Excellence in Molecular Cell Science, Chinese Academy of Sciences, University of Chinese Academy of Sciences, Shanghai, China. [2] School of Life Science and Technology, ShanghaiTech University, Shanghai, China. [3] School of Life Science, Hangzhou Institute for Advanced Study, University of Chinese Academy of Sciences, Hangzhou, China. [4] These authors contributed equally: Wenjing Li, Tianlong Zhang, Mingliang Sun. ✉email: jpding@sibcb.ac.cn

Covalent modifications of DNA bases, such as $C^5$-methyl-cytosine (5mC), $N^4$-methylcytosine (4mC), and $N^6$-methyladenine (6mA), were discovered to exist in many living organisms. Among these modifications, 5mC is the most common and well-studied DNA modification and an important epigenetic mark in eukaryotes, which plays diverse functional roles in many fundamental cellular and developmental processes, including transcription, chromatin remodeling, X-chromosome inactivation, genomic imprinting, pluripotency of stem cells, embryonic development, and tumor development[1–7]. In the past two decades, there is increasing interest in the discovery of additional DNA base modifications and their biological functions in the epigenetics field. It was recently found that 5mC in DNA can be oxidized stepwise into $C^5$-hydroxymethyl-cytosine (5hmC), $C^5$-formyl-cytosine (5fC), and $C^5$-carboxyl-cytosine (5caC) by the ten–eleven translocation (TET) enzymes, using $Fe^{2+}$ as cofactor and 2-oxoglutarate (2-OG) as co-substrate[8–10]. These 5mC derivatives are also considered as stable epigenetic marks and are demonstrated to play vital roles in the epigenetic regulation of gene expression[2–4,7,11,12]. In addition, two additional DNA base modifications, 4mC and 6mA, which were previously thought to be restricted to prokaryotes, were recently reported to be present in the genomic DNAs of a variety of Metazoa[13,14]. However, the existence of 4mC and 6mA in Metazoa is still in debate: some studies suggest that the 6mA modification might be a potential epigenetic mark and plays important roles in gene transcription and chromatin remodeling[15–21]; others suggest that the detected 6mA and 4mC might be due to artifacts in measurements[22].

The TET proteins responsible for the oxidation of DNA 5mC constitute a subfamily of the $Fe^{2+}$/2-OG-dependent dioxygenases that catalyze diverse oxidative reactions and play vital roles in many biological processes[7,12,23–25]. The majority of the $Fe^{2+}$/2-OG-dependent dioxygenases utilize $Fe^{2+}$ as cofactor, and 2-OG and $O_2$ as co-substrates, producing $CO_2$ and succinate as co-products. These enzymes are structurally characterized by a conserved double-stranded β-helix (DSBH) core fold consisting of a major and a minor β-sheet with the more open end containing the $Fe^{2+}$ and 2-OG binding sites. The DSBH core is usually supplemented with additional β-strands, α-helices, or domains at the N- or/and C-termini, which play roles in stabilization of the DSBH fold, and in recognition and binding of the substrate. The catalytically required $Fe^{2+}$ is usually chelated by three highly conserved residues of the conserved HXD/E…H motif; however, the residues involved in the 2-OG binding are not universally conserved, but are subfamily characteristic[23–25].

Very lately, a TET homologue in a green alga *Chlamydomonas reinhardtii*, named as CMD1, was found to exhibit a novel enzymatic activity[26]. In vitro biochemical studies showed that recombinant CMD1 can catalyze the conjugation of a glyceryl moiety to the $C^5$-methyl group of 5mC via a direct carbon–carbon bond, leading to a new DNA base modification named as $C^5$-glyceryl-methylcytosine (5gmC). Like the other TET proteins, the enzymatic activity of CMD1 is $Fe^{2+}$-dependent and requires the existence of a conserved HXD…H motif. However, unlike the canonical TET proteins, CMD1 uses vitamin C (VC, L-ascorbic acid) rather than 2-OG as co-substrate, which surprisingly contributes the glyceryl moiety (the C4–C6 atoms) to the $C^5$-methyl group of 5mC with glyoxylate and $CO_2$ as co-products. VC is widely demonstrated to function as a potent antioxidant to reduce and neutralize oxidizing agents, such as free radicals and reactive oxygen species or as a reducing agent for $Fe^{2+}$/2-OG-dependent dioxygenases to enhance their catalytic activities[27–35]; but no studies had shown that VC could act as a co-substrate. In vivo functional studies further showed that the 5gmC modification exists in the green alga genomic DNA and the modification

level is substantially reduced when *CMD1* was knocked out, and that the 5gmC modification plays a vital role in the regulation of photosynthesis for adaptation under intense light[26]. These studies not only identified a new eukaryotic DNA base modification as a potential epigenetic mark and suggested a new fate of 5mC in the green alga genome, but also uncovered a novel activity of a TET-like enzyme and revealed a functional role of VC as co-substrate in the epigenetic regulation. However, the underlying molecular mechanism remains unknown.

In this work, we performed in vitro DNA binding assay for CMD1 and showed that CMD1 exhibits comparable binding affinities for DNAs of different lengths, structures, and 5mC levels, and displays a moderate substrate preference for 5mCpG-containing DNA than 5mCpC-, 5mCpA-, and 5mCpT-containing DNAs. We determined the crystal structures of CMD1 in apo form and in complexes with VC or dsDNA, and with both VC and dsDNA. Structural analysis shows that CMD1 assumes a typical DSBH fold. The $Fe^{2+}$ at the active site is coordinated by the three conserved residues of the HXD…H motif. The lactone form of VC binds to the active site and mono-coordinates the $Fe^{2+}$ in a manner different from 2-OG in the $Fe^{2+}$/2-OG-dependent dioxygenases. The dsDNA binds to a positively charged cleft of CMD1 via the phosphate backbone, and the methylcytosine or cytosine is flipped out of the dsDNA, and inserted into the active site and is recognized by the enzyme in a manner similar to the TET proteins. In vitro enzymatic activity assay shows that mutations of the key residues involved in the binding of $Fe^{2+}$ and VC abolish the activity, and those involved in the DNA binding significantly impair the activity. Our structural and biochemical data together reveal the molecular basis for how CMD1 recognizes the DNA substrate and utilizes VC as the co-substrate, and provide mechanistic insight into the catalytic reaction of the 5gmC DNA modification by CMD1.

## Results

**Biochemical characterization of CMD1.** The full-length CMD1 (residues 1–532) used in the biochemical studies and the MBP-fused CMD1 used in the structural studies were expressed in *Escherichia coli*, and purified using a combination of affinity chromatography and gel filtration chromatography (Supplementary Fig. 1a, b). The previous biochemical study showed that CMD1 can catalyze the formation of 5gmC from 5mC in DNA using VC as co-substrate in the presence of ATP (1 mM)[26]. In the optimization of the activity assay, we found that CMD1 exhibits a much higher activity in the absence than in the presence of ATP (Supplementary Fig. 1c, d). Thus, ATP was removed from the reaction mixture in the activity assay of CMD1 in this work. As the DNA substrate (0.5–1.1 kb) used in the activity assay in the previous study is too long for structural study, we analyzed the binding affinities of CMD1 for a variety of DNAs with different lengths, sequences, and structures using an in vitro bio-layer interferometry (BLI) assay to investigate the substrate specificity of CMD1 and to identify suitable DNA substrate for co-crystallization. The genomic DNA of *C. reinhardtii* is GC-rich (nearly 65%) with a relatively low 5mC level (~0.4%)[36,37]. To examine whether CMD1 exhibits preference for DNA substrates with GC-rich and/or different 5mC levels, we first measured the binding of CMD1 with four dsDNAs (40 bp) containing unmodified C, two of which are AT-rich (dsDNA1 and dsDNA2) and two are GC-rich (dsDNA3 and dsDNA4), and the results show that the binding affinities ($K_D$) of CMD1 with all these dsDNAs are in the range of 1.97–2.50 μM (Supplementary Table 1), indicating that CMD1 exhibits no notable preference for the sequence of dsDNA (GC-rich or AT-rich). Then, we measured the binding of CMD1 for dsDNAs (40 bp) containing different

5mC levels (dsDNA5 and dsDNA6), and found that the binding affinities of CMD1 for dsDNAs containing one or two 5mC modifications are ~2.28–2.52 μM, which are comparable to those for the unmethylated dsDNAs (dsDNA1–4) (Supplementary Table 1), indicating that CMD1 also exhibits no notable preference for the 5mC levels. In addition, we measured the binding of CMD1 toward dsDNAs of different lengths (10–30 bp; dsDNA7, dsDNA8, and dsDNA9), and found that the binding affinities of CMD1 for these shorter dsDNAs (1.74–2.15 μM) are comparable to those for the 40-bp dsDNAs (dsDNA1–4; Supplementary Table 1). Moreover, we measured the binding of CMD1 toward a 40-nt ssDNA, and found that the binding affinity of CMD1 for this ssDNA is 2.77 μM, which is comparable to those for the dsDNAs (Supplementary Table 1). Furthermore, we measured the binding affinities of CMD1 toward several DNAs with different structures, including Y-DNA, bubble, 5′-flap, 3′-flap, 5′-OH, and 3′-OH, and found that the binding affinities of CMD1 for all these DNAs are in the range of 1.19–1.61 μM, which are slightly stronger than those for the dsDNAs and ssDNA examined in this study (Supplementary Table 1), indicating that CMD1 has insignificant preference for specific DNA structures. Taken together, our in vitro binding assay results show that CMD1 exhibits comparable binding affinities for DNA substrates of different lengths, structures, and 5mC levels.

It was reported previously that human TET2 exhibits a strong substrate preference for 5mCpG-containing DNA than 5mCpC- and 5mCpA-containing DNAs[38]. To investigate whether 5mC in different contexts would affect the activity of CMD1, we carried out enzymatic activity assay using 90-bp dsDNAs containing six 5mCpX sites, where X is G, A, C, or T (Supplementary Table 2). The results show that CMD1 exhibits the highest activity on 5mCpG-containing DNA, relatively lower activity (~50%) on 5mCpC- and 5mCpA-containing DNAs, and a much lower activity (~15%) on 5mCpT-containing DNA (Supplementary Fig. 1e), suggesting that CMD1 has a moderate substrate preference for 5mCpG-containing DNA and could accommodate the substitution of guanine by adenine or cytosine to some extent.

**Crystal structure of CMD1 in apo form.** Crystalization of the full-length CMD1 in the absence and presence of VC or/and DNA substrates of different lengths, sequences, and structures did not yield any crystals. After various trails, an N-terminal MBP-fused CMD1 with an α-helical linker yielded crystals of high diffraction quality, which led to the successful structure determination of CMD1 in apo form. The structure of the MBP-fused CMD1 in apo form was solved by the single-wavelength anomalous dispersion (SAD) method at 2.20 Å resolution with the asymmetric unit containing one MBP–CMD1 (Table 1). In the structure, most residues of the catalytic domain of CMD1 and the MBP are defined with unambiguous electron density except for a few surface exposed regions (residues 177–184 and 244–249 of CMD1, and residues 1–7 and 55–57 of MBP); however, the C-terminal region of CMD1 (residues 509–532) is completely disordered (Supplementary Table 3). SDS–PAGE analysis shows that the MBP-fused CMD1 are stable in the storage buffer; however, the C-terminal region of the MBP-fused CMD1 was partially degraded in the crystals and completely degraded in the crystallization solution (Supplementary Fig. 1f).

Similar to other Fe$^{2+}$/2-OG-dependent dioxygenases[24,25,39], the catalytic domain of CMD1 contains a typical DSBH fold (Fig. 1a and Supplementary Fig. 2). Compared to the common DSBH fold in other dioxygenases, the DSBH fold of CMD1 is much larger with the major β-sheet consisting of ten β-strands (β5–β6, β8, β10, β11–β13, β15, β19, and β22) and the minor β-sheet consisting of five β-strands (β7, β14, β17–β18, and β20).

The DSBH fold packs with two α-helices (α4 and α5) on the outer surface of the minor β-sheet and ten α-helices (α1–α3 and α6–α12) on the outer surface of the major β-sheet to form a four-layered architecture. In addition, there are six β-strands (β1–β2, β3–β4, β16, and β21) which form another β-sheet to flank one vertical side of the DSBH fold, and two short β-strands (β9 and β23) to fold along the N-terminal part of the β10 strand as an extension of the major β-sheet (Fig. 1a and Supplementary Fig. 2). There is a metal ion bound at the active site, which is interpreted as Fe$^{2+}$ owing to the presence of Fe$^{2+}$ in the protein solution and a reasonable $B$-factor for the Fe$^{2+}$ in the refinement (Table 1). Indeed, the ICP-OES (inductively coupled plasma optical emission spectrometer) analysis shows that Fe is the most abundant metal in the protein solution without or with supplementation of $(NH_4)_2Fe(SO_4)_2$ (Supplementary Table 4), further supporting that the bound metal ion at the active site is an iron ion.

The MBP was fused to the N-terminus of CMD1 via an α-helical linker to facilitate crystal packing, which results in the formation of a long α-helix comprising the last α-helix of MBP, the α-helical linker, and the α1 helix of CMD1 (Supplementary Fig. 3a). In the apo CMD1 structure, the MBP is located in adjacent to CMD1, and its position and orientation are dictated by the long α-helix. There are only a few interactions between one short α-helix (residues 44–57) from the N-domain of MBP and two surface loops of CMD1 (the β8–β9 and α12–β23 loops; Supplementary Fig. 3a). Crystal packing is mediated mainly by several α-helices of CMD1 (α2 and α5–α8) supplemented by the MBP (Supplementary Fig. 3b). As the MBP is located distantly from the active site of CMD1, it has no direct impact on the chemical property and structure of the active site.

**Crystal structures of CMD1 in complexes with VC, DNA, and both VC and DNA.** Co-crystallization of the MBP-fused CMD1 with VC yielded crystals of CMD1 in apo form. ITC (Isothermal Titration Calorimetry), SPR (Surface Plasmon Resonance), MST (Micro Scale Thermophoresis), and BLI analyses showed undetectable binding of VC to CMD1, indicating that CMD1 has a weak binding affinity for VC. Eventually, the crystals of CMD1 in complex with VC were obtained by soaking the apo CMD1 crystals in the reservoir solution containing VC, and the structure was solved by the molecular replacement (MR) method at 2.20 Å resolution (Table 1). After many co-crystallization trails of the MBP-fused CMD1 with various DNA substrates of different lengths, sequences, and structures, we obtained crystals of CMD1 in complexes with several dsDNAs with or without 3′-overhang (9–16 nt). Among these CMD1–DNA complexes, the crystals of CMD1 in complex with a 14-nt DNA which comprises a 10-bp dsDNA region and a 4-nt 3′-overhang with or without a 5mC at the 3-position on both the forward (or substrate) strand and the reverse (or non-substrate) strand diffracted X-rays to relatively higher resolution. The structures of the CMD1 in complexes with 5mC–DNA or DNA were solved by the MR method at 2.10 and 2.15 Å resolution, respectively (Table 1). The crystals of CMD1 in complex with both DNA and VC were obtained by soaking the crystals of the CMD1–DNA complex in the reservoir solution containing VC, and the structure was solved by the MR method at 2.40 Å resolution (Table 1 and Fig. 1b). In all of the structures, the HXD…H motif and the metal ion are well-defined with evident electron density; in all of the complex structures, the co-substrate and/or a large portion of the DNA substrate are also well-defined with evident electron density (Fig. 1c and Supplementary Fig. 4a–e).

In the structure of the CMD1–5mC–DNA–VC complex, the electron density clearly shows that the bound VC is in the lactone

**Table 1 Summary of diffraction data and refinement statistics.**

|  | Apo | VC | 5mC–DNA | DNA | 5mC–DNA + VC | Se-Met |
|---|---|---|---|---|---|---|
| PDB code | 7CY4 | 7CY5 | 7CY6 | 7CY7 | 7CY8 |  |
| **Data collection** |  |  |  |  |  |  |
| Beamline | BL19U1 | BL19U1 | BL19U1 | BL19U1 | BL18U | BL17U1 |
| Space group | C2 | C2 | C2 | C2 | C2 | C2 |
| Wavelength (Å) | 0.9785 | 0.9785 | 0.9785 | 0.9785 | 0.9792 | 0.9792 |
| Cell dimensions |  |  |  |  |  |  |
| $a$ (Å) | 154.7 | 153.8 | 153.2 | 153.6 | 154.5 | 154.7 |
| $b$ (Å) | 125.6 | 126.4 | 127.1 | 125.8 | 125.7 | 122.5 |
| $c$ (Å) | 64.2 | 64.2 | 64.2 | 64.0 | 64.1 | 64.2 |
| $\beta$ (°) | 103.2 | 102.3 | 102.7 | 102.8 | 103.0 | 102.9 |
| Resolution (Å) | 47.77–2.20 | 48.38–2.20 | 48.39–2.10 | 49.28–2.15 | 49.28–2.40 | 50.00–2.50 |
|  | (2.26–2.20)[a] | (2.25–2.20) | (2.15–2.10) | (2.21–2.15) | (2.50–2.40) | (2.59–2.50) |
| Observed reflections | 337,182 | 271,911 | 420,327 | 400,272 | 165,381 | 271,204 |
| Unique reflections | 48,745 | 55,672 | 62,383 | 60,566 | 35,155 | 40,382 |
|  | (2437) | (2783) | (3119) | (3028) | (1758) | (3987) |
| $R_{merge}$ (%)[b] | 7.3 (123.6) | 8.4 (102.9) | 6.9 (84.1) | 6.7 (52.1) | 8.8 (101.8) | 8.1 (60.6) |
| $I/\sigma(I)$ | 17.2 (1.7) | 16.3 (1.8) | 18.6 (2.2) | 15.8 (2.0) | 13.1 (1.6) | 21.6 (2.8) |
| Spherical completeness (%) | 81.3 (46.5) | 91.7 (71.0) | 88.6 (56.2) | 94.0 (62.1) | 75.3 (31.5) | 99.6 (98.3) |
| Ellipsoidal completeness (%) | 94.3 (91.6) | 95.5 (97.6) | 96.6 (92.0) | 96.7 (82.4) | 94.6 (86.2) | — |
| Redundancy | 6.9 (6.6) | 4.9 (5.1) | 6.7 (6.7) | 6.6 (3.3) | 4.7 (4.5) | 6.7 (6.3) |
| $CC_{1/2}$ | 1.00 (0.56) | 1.00 (0.65) | 1.00 (0.78) | 1.00 (0.82) | 1.00 (0.53) | 1.00 (0.58) |
| **Refinement** |  |  |  |  |  |  |
| Resolution (Å) | 47.77–2.20 | 48.38–2.20 | 48.39–2.10 | 49.28–2.15 | 49.28–2.40 |  |
| No. of reflections |  |  |  |  |  |  |
| Working set | 45,643 | 53,453 | 59,114 | 57,529 | 33,382 |  |
| Test set | 1996 | 2000 | 2987 | 3035 | 1757 |  |
| $R_{work}/R_{free}$ (%)[c] | 17.9/22.3 | 18.1/22.1 | 18.2/22.7 | 18.8/22.1 | 19.0/23.5 |  |
| No. atoms |  |  |  |  |  |  |
| Macromolecules | 6521 | 6575 | 6993 | 6969 | 6718 |  |
| Ligand | — | 12 | — | — | 12 |  |
| Ion | 1 | 1 | 1 | 1 | 1 |  |
| Solvent | 236 | 428 | 520 | 438 | 210 |  |
| Wilson $B$-factors (Å$^2$) | 39.0 | 37.2 | 28.4 | 31.1 | 41.9 |  |
| $B$-factors |  |  |  |  |  |  |
| Macromolecules | 54.8 | 38.4 | 35.4 | 41.6 | 52.5 |  |
| Ligand | — | 48.5 | — | — | 49.5 |  |
| Ion | 66.0 | 81.6 | 36.0 | 44.4 | 91.9 |  |
| Solvent | 52.9 | 40.1 | 46.8 | 39.1 | 43.0 |  |
| RMS deviations |  |  |  |  |  |  |
| Bond lengths (Å) | 0.009 | 0.010 | 0.007 | 0.006 | 0.008 |  |
| Bond angles (°) | 1.23 | 1.27 | 1.18 | 1.12 | 1.28 |  |
| Ramachandran plot (%) |  |  |  |  |  |  |
| Favored | 97.8 | 97.7 | 97.5 | 97.4 | 97.4 |  |
| Allowed | 2.2 | 2.3 | 2.5 | 2.6 | 2.6 |  |
| Outliers | 0.0 | 0.0 | 0.0 | 0.0 | 0.0 |  |

[a]Numbers in parentheses represent the highest resolution shell.
[b]$R_{merge} = \sum_{hkl}\sum_i |I_i(hkl) - \langle I(hkl)\rangle| / \sum_{hkl}\sum_i I_i(hkl)$.
[c]$R = \sum_{hkl}||F_o| - |F_c|| / \sum_{hkl}|F_o|$.

form, suggesting that the catalytic reaction does not take place in the crystals or crystallization drops. To explore the possible reason (s), we analyzed the buffer and pH profile of the catalytic reaction. The results show that CMD1 exhibits substantially differed activity in different buffers with different pH values (Supplementary Fig. 4f). Among the conditions examined, CMD1 displays the highest activity in the HEPES buffer at pH 7.0 which was used in the enzymatic activity assay; however, it is inactive in the citrate buffers at pH 5.0–6.0, which were used in the crystallization solutions for both the apo CMD1 and the CMD1–DNA complex. This might explain why CMD1 is inactive in the crystals and the crystallization drops. The molecular basis for the CMD1 inactivation at the acidic conditions is unclear. It is possible that the acidic conditions might affect the protonation states of VC and/or residues involved in the

binding of the metal ion, co-substrate and substrate, and consequently the catalytic reaction.

Comparison of the apo and VC-bound CMD1 structures shows that the VC binding does not cause notable conformational changes at the active site, and in the overall structure of CMD1 (RMSD of 0.12 Å for 492 Cα atoms) (Fig. 1d and Supplementary Table 5). Comparison of the apo and DNA-bound CMD1 structures shows that the DNA binding also induces no notable conformational changes in the overall structure of CMD1 (RMSD of 0.24 Å for 494 Cα atoms; Fig. 1d and Supplementary Table 5). However, a disordered region adjacent to the active site (residues 244–249, the β11–β12 loop) in the apo CMD1 structure is well-defined in the CMD1–DNA structures and is involved in the DNA binding and recognition (Fig. 1d). In addition, structural

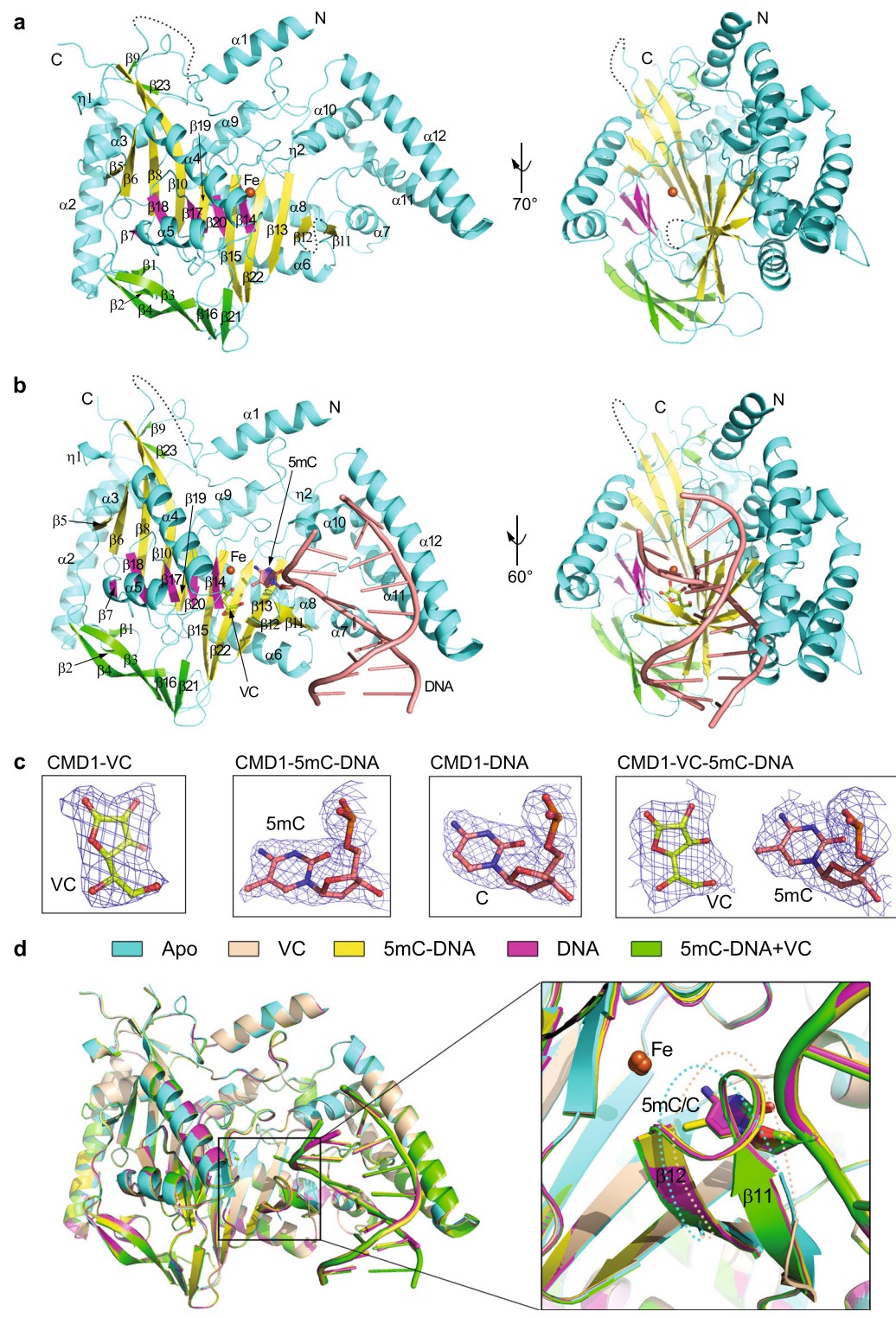

comparison shows that the overall structure of CMD1 in the CMD1–5mC–DNA–VC complex exhibits no notable conformational differences from that in the CMD1–VC and CMD1–DNA complexes (Fig. 1d). As the structure of the CMD1–5mC–DNA–VC complex represents the most biologically relevant state, we will use it in the following structural analyses unless otherwise specified.

**The active site of CMD1.** The active site of CMD1 is located on the more open end of the DSBH fold and consists of the binding sites for the $Fe^{2+}$, the VC and the flipped base of DNA substrate (Fig. 1a, b). In the apo CMD1 structure, the $Fe^{2+}$ is coordinated by three strictly conserved residues (His345, Asp347, and His397) from the characteristic HXD/E…H motif of the $Fe^{2+}$/2-OG-dependent dioxygenases[24,25,39], and three water molecules with an octahedral geometry (Fig. 2a). In the CMD1–5mC–DNA–VC structure, the $Fe^{2+}$ also makes interactions with the C2-hydroxyl group of VC and Wat3 with a tetrahedral pyramidal coordination geometry; Wat2 is replaced by the side chain of Arg244 (Fig. 2b). In the other CMD1 structures, the $Fe^{2+}$ maintains the

**Fig. 1 Overall structure of CMD1. a** Ribbon representation of CMD1 in apo form. The minor β-sheet and major β-sheet of the DSBH fold are colored in magenta and yellow, respectively. The two layers of α-helices are colored in cyan. The extra β-strands flanking the minor and major β-sheet of the DSBH fold are colored in green. The $Fe^{2+}$ at the active site is shown with an iron-red sphere. The disordered regions are indicated with black dotted lines. The secondary structure elements are labeled. The N- and C-termini are indicated. **b** Ribbon representation of CMD1 in complex with VC and 5mC–DNA. The color coding for CMD1 is the same as **a**. The bound dsDNA is colored in salmon and the 5mC base is shown with a ball-and-stick model. The VC is shown as a ball-and-stick model and colored in lemon. **c** Representative simulated annealing composite omit maps (contoured at 1.0 σ level) for the VC in the CMD1–VC complex, the flipped-out 5mC at the 3-position of the substrate strand in the CMD1–5mC–DNA complex, the flipped-out C at the 3-position of the substrate strand in the CMD1–DNA complex, and the VC and the flipped-out 5mC at the 3-position of the substrate strand in the CMD1–5mC–DNA–VC complex. **d** Comparison of the structures of CMD1 in apo form (cyan), in complex with VC (wheat), in complex with 5mC–DNA (yellow), in complex with DNA (magenta), and in complex with 5mC–DNA and VC (green). The zoom-in window shows that the disordered β11–β12 loop in the structures of CMD1 in the apo form and VC-bound complex is well-defined in the structures of CMD1 in the DNA-bound complexes. The disordered β11–β12 loop in the apo form and VC-bound complex are indicated with dotted lines.

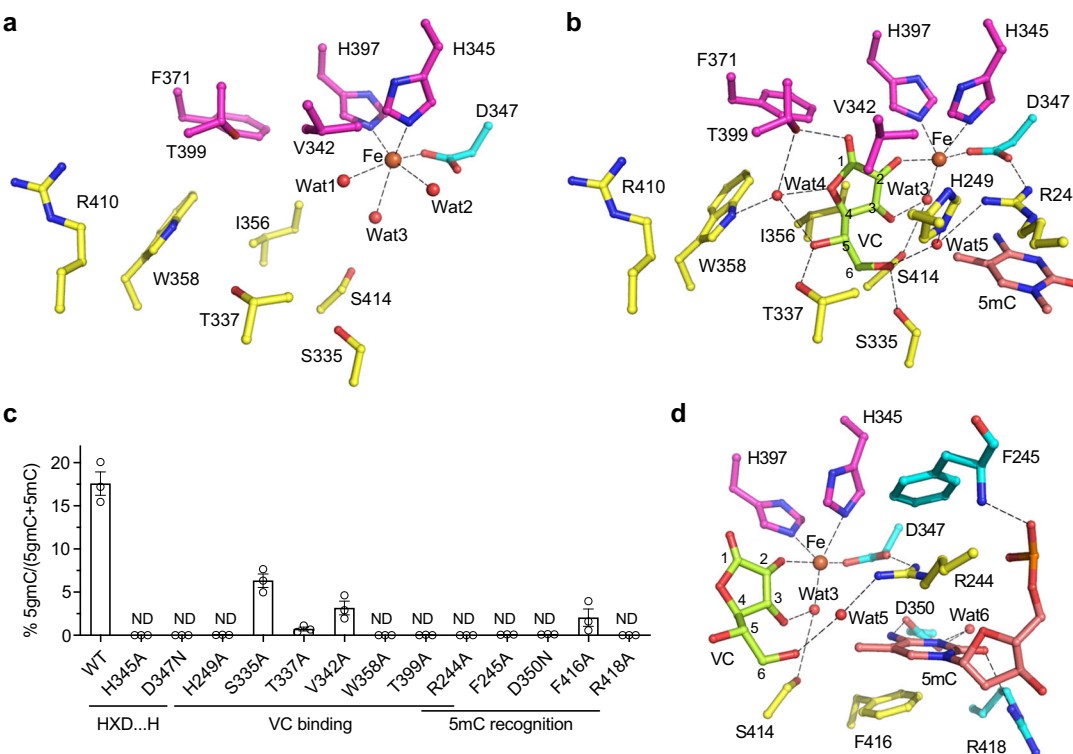

**Fig. 2 Structure of the active site. a** Structure of the active site of CMD1 in apo form showing the interactions of the $Fe^{2+}$ with the surrounding residues. **b** Structure of the active site of CMD1 in complex with VC and 5mC–DNA showing the interactions of the $Fe^{2+}$ and the VC, with the surrounding residues. The carbon atoms of VC are numbered. **c** Effect of mutations on the enzymatic activity of CMD1. Data are presented as mean values ± SEM (standard error of the mean). $n = 3$ independent replicates. **d** Structure of the active site of CMD1 in complex with 5mC–DNA and VC showing the interactions of the $Fe^{2+}$ and the flipped-out 5mC base, with the surrounding residues and VC. The hydrogen bonds and coordination bonds are indicated with black dotted lines.

coordination bonds with the three conserved residues (Supplementary Fig. 5a–c). As expected, the CMD1 mutants containing mutations H345A and D347N completely lost the activity because these mutations would impair the $Fe^{2+}$ binding (Fig. 2c). Of note, the CMD1 mutant containing mutation H397A could not be expressed for unknown reason.

In the CMD1–5mC–DNA–VC complex, the VC forms a number of direct and indirect hydrogen-bonding interactions with the surrounding residues, including Arg244, His249, Ser335, Thr337, Trp358, Thr399, and Ser414. Besides, the C2-hydroxyl of VC forms a coordination bond with the $Fe^{2+}$. In addition, the VC also forms hydrophobic interactions with the side chains of Val342, Ile356, and Phe371 (Fig. 2b). In the CMD1–VC complex, the VC makes essentially identical interactions with the protein (Supplementary Fig. 5a), suggesting that the DNA binding has no direct impact on the VC binding. To verify the functional roles of the key residues involved in the VC binding, we performed

mutagenesis and activity assay. Mutations of Arg244, His249, Thr337, Trp358, and Thr399 to Ala abolish the activity of CMD1, and mutations of Ser335 and Val342 reduce the activity by two to threefolds (Fig. 2c), indicating that the hydrophilic interactions play a major role in the VC binding, and the activity of CMD1 is dependent on the proper binding of VC. Of note, the $His_6$-SUMO tag of the S414A mutant could not be efficiently cleaved for unknown reason, and thus the $His_6$-SUMO tagged S414A mutant was not used in the activity assay.

In the CMD1–5mC–DNA–VC complex, the 5mC base of the substrate strand is flipped out of the dsDNA and inserted into the active site of CMD1, and is specifically recognized by several residues of CMD1 (Fig. 2d). In detail, the N4 and O2 atoms of the 5mC base make hydrogen-bonding interactions with the side chains of Asp350 and Arg418, respectively. The N3 atom of the 5mC base forms an indirect hydrogen bond with the side chain of Asp350 via a water molecule. The pyrimidine ring of the 5mC is

sandwiched between the side chains of Arg244 and Phe416 via cation–π and π–π interaction, respectively. The side chain of Arg244 is further stabilized via a cation–π interaction with the side chain of Phe245 and a hydrogen-bonding interaction with the side chain of Asp347. In the CMD1–5mC–DNA complex, the 5mC base makes almost identical interactions with the protein (Supplementary Fig. 5b). Intriguingly, although the C5-methyl group of the 5mC base points to the active site, it is not specifically recognized by any residue (Fig. 2d and Supplementary Fig. 5b). This suggests that an unmethylated C base could bind to the active site of CMD1 as well. Indeed, in the CMD1–DNA complex containing no 5mC modification at the 3-position of the DNA, the unmodified C base is also flipped out of the dsDNA and inserted into the active site, which makes similar interactions with the surrounding residues (Supplementary Fig. 5c). These results are consistent with our biochemical data showing that CMD1 exhibits no notable preference for 5mC modification (Supplementary Table 1). On the other hand, docking of other bases into the active site of CMD1 shows that the O4 atom of the T base could not form a favorable hydrogen-bonding interaction with the side chain of Asp350, and the A and G bases would have steric conflicts with the side chains of Asp350 and Arg244, and could not from hydrogen bonds with the side chain of Arg418 (Supplementary Fig. 5d). These results explain why only 5mC or C can bind to the active site of CMD1. To verify the functional roles of the key residues involved in the 5mC binding, we performed mutagenesis and activity assay. The results show that mutations of these residues abolish or significantly impair the CMD1 activity (Fig. 2c). Specifically, mutations of Arg244, Phe245, and Arg418 to Ala and Asp350 to Asn result in a complete loss of the CMD1 activity, and mutation of Phe416 to Ala lead to eightfold reduction of the activity (Fig. 2c). Taken together, our structural and biochemical results suggest that residues Arg244, Phe245, Asp350, and Arg418 play vital roles and residue Phe416 plays an important role in the binding and recognition of the 5mC base.

It was previously reported that TET could oxidize thymine, which contains a C5-methyl group on the pyrimidine scaffold to 5-hydroxymethyluracil in mouse embryonic stem cells[40]. Thus, we performed enzymatic activity assays of the wild-type CMD1 and the D350N mutant (which might be able to form a hydrogen bond with the O4 atom of the T base) toward 0.5-kb 5mCpG- and CpG-containing DNAs (which contain ~25.5% T base) to examine whether CMD1 could catalyze the glyceryl transfer from VC to T. The results showed that for the 5mCpG-DNA substrate, the wild-type CMD1 could catalyze the formation of 5gmC efficiently and the D350N mutant could generate a very little amount of 5gmC; and for the CpG-DNA substrate, both the wild-type and mutant CMD1 cannot catalyze the formation of 5gmC. On the other hand, in all of these assays, no new product peak could be identified at the expected position for $C^5$-glyceryl-thymine (5gT or 5gmU) and the abundance of the T peak has no notable change (Supplementary Fig. 5e). These results suggest that the wild-type and the D350N mutant CMD1 cannot catalyze the reaction of a glyceryl transfer from VC to T.

**Binding and recognition of DNA.** In the CMD1–5mC–DNA–VC complex, a major portion of the dsDNA region (nucleotides 1–8 of the substrate strand and nucleotides 3–11 of the non-substrate strand), including the 5mC at the 3-position of the substrate strand is clearly defined in the electron density map, whereas the 3′-overhang regions are largely disordered (Fig. 1b, c and Supplementary Fig. 4e). The DNA binds to a shallow and positively charged surface cleft formed mainly by helices α7 and α10–α12 of CMD1 (Figs. 1b and 3a). CMD1 interacts mainly with the phosphate backbone of the DNA via extensive hydrophilic inter-actions (Fig. 3b–d). For the substrate strand, the phosphate group of 5mC forms a hydrogen bond with the main-chain amine of Phe245; the phosphate group of G4 forms a hydrogen bond with the side chain of Tyr270 and a salt bridge with the side chain of Arg418; the phosphate group of C5 forms a direct hydrogen bond with the main-chain amine of Ala271 and indirect hydrogen bonds with the side chain of Asp262 and the main-chain amine of Ala271 via a water molecule and the main-chain amine of Val272 via another water molecule; and the phosphate group of G6 forms two direct salt bridges with the side chains of Lys264 and Arg275 (Fig. 3b, d). For the non-substrate strand, the ribose moiety of C7′ forms a hydrogen bond with the side chain of Arg461; the phosphate group of G9′ forms a hydrogen bond with the side chain of Ser465; the phosphate group of G10′ forms a salt bridge with the side chain of Arg471, a direct hydrogen bond with the main-chain carbonyl of Ala468, and indirect hydrogen bonds with the side chains of Arg471 and Asn260 and the main-chain amine of Lys472 via a water molecule (Fig. 3c, d). In addition, the side chain of Asn261 forms direct hydrogen bonds with the guanine moiety of the unpaired G8′ and the ribose moiety of G9′. In the CMD1–DNA complexes with or without 5mC modification, a similar portion of the dsDNA region (nucleotides 1–7 of the substrate strand and nucleotides 3–11 of the non-substrate strand), including the modified or unmodified C3 of the substrate strand is also clearly defined in the electron density maps (Supplementary Fig. 4c, d), and the DNA makes essentially identical interactions with the protein except that the phosphate group of A11′ also forms a salt bridge with the side chain of Lys479 probably due to the relatively higher resolution of the CMD1–DNA complexes (Supplementary Fig. 6a, b). These results also suggest that the VC binding has no direct impact on the DNA binding. Furthermore, structural analysis reveals that there are several basic residues located adjacent to the DNA-binding site, including Lys420, Arg484, and Arg494 (Fig. 3e). Although these residues have no interactions with the DNA in the CMD1–5mC–DNA–VC and CMD1–DNA complexes, a modeling study suggests that these residues could make interactions with a 20-bp dsDNA modeled based on the extension of the dsDNA region of the bound 14-nt DNA (Supplementary Fig. 6c).

To verify the functional roles of the key residues involved in the DNA binding, we performed mutagenesis and biochemical analyses. In vitro binding assays show that mutations of these residues have only slight or insignificant impacts on the binding affinity of CMD1 toward the DNA substrate (0.3–1.5-folds; Fig. 3f), consistent with the structural data showing that a number of residues of CMD1 make extensive hydrophilic interactions with the DNA and single mutation of any of these residues could not disrupt the DNA binding. On the other hand, the enzymatic activity assays show that mutations of these residues abolish or significantly impair the CMD1 activity (Fig. 3f). Specifically, mutations of Tyr270, Ser465, and Arg471 abolish the activity; mutations of Arg275 and Lys472 lead to >30-fold reduction of the activity; and mutations of Lys264, Asn261, and Arg461 lead to about five to eightfold reduction of the activity. It is likely that mutations of these residues might affect the interactions of CMD1 with the DNA, and thus the precise positioning of the 5mC base at the active site, leading to the abolishment or impairment of the activity. In addition, mutations of the basic residues adjacent to the DNA-binding site (Lys420, Arg484, and Arg494) result in >30-fold decrease of the activity (Fig. 3f). This is consistent with our modeling study showing that these residues might be involved in the binding of a longer DNA (Supplementary Fig. 6c). Taken together, residues Tyr270, Arg275, Ser465, Arg471, and Lys472 play vital roles and residues Lys264, Asn261, and Arg461 play important roles in the binding and proper positioning of the DNA. Of note, the His₆-SUMO tag of the K479A mutant could

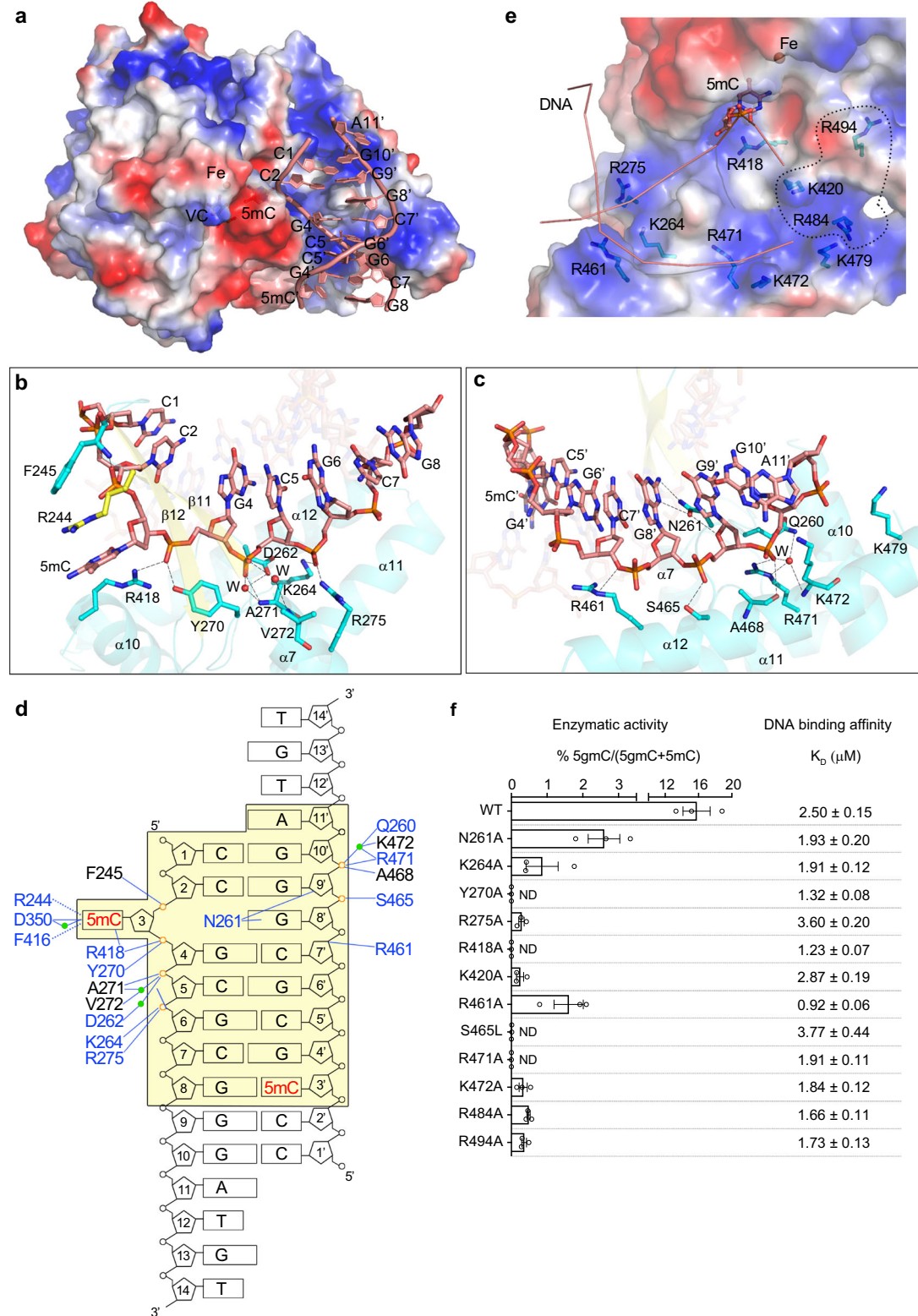

not be efficiently cleaved for unknown reason, and thus the His$_6$-SUMO-tagged K479A mutant was not used in the activity and DNA binding assays.

In all of the DNA-bound CMD1 structures, there is no direct interactions between the DNA and the MBP. However, nucleotide A11′ of the reverse chain points toward the MBP with a distance of ~12 Å (between the ribose of A11′ and Asn13 of the MBP; Supplementary Fig. 6c). Modeling studies indicate that the 3′ end

of the non-substrate strand of the 14-nt DNA could make interaction with the MBP, and this interaction might stabilize the binding of the 14-nt DNA with the MBP-fused CMD1; and that a longer overhang or a longer dsDNA would have steric conflict with the MBP (Supplementary Fig. 6c). These results are in agreement with our biochemical data showing that the MBP-fused CMD1 exhibits a slightly higher binding affinity (0.81 ± 0.03 μM) toward the 14-nt DNA used in the structural study than

**Fig. 3 Interactions between CMD1 and DNA in the CMD1–5mC–DNA–VC structure. a** Electrostatic potential surface of CMD1. Bases of the DNA defined in the structure are shown in ring representation and labeled. The DNA-binding site of CMD1 is largely positively charged (blue). **b, c** Detailed interactions of CMD1 with the substrate strand (**b**) and the non-substrate strand (**c**) of the bound DNA. The bound DNA and the residues of CMD1 involved in the interactions are shown with ball-and-stick models. Water molecules are shown in red spheres. The color coding is the same as Fig. 1b. The hydrogen bonds are indicated with black dotted lines. **d** A schematic diagram showing the interactions between the CMD1 and the 5mC–DNA in the CMD1–5mC–DNA–VC complex. The defined region of the DNA is highlighted in a shaded yellow box. The hydrophilic interactions are indicated with blue solid lines, and the hydrophobic, cation–π, and π–π stacking interactions with blue dashed lines. The phosphates of the DNA involved in the interactions with CMD1 are highlighted in orange. The residues of CMD1 interacting with the DNA via the side chains and main chains are colored in blue and black, respectively. Water molecules are indicated with green spheres. **e** Structure of the DNA-binding site of CMD1. The bound DNA is shown with a ribbon representation and the flipped-out 5mC base is shown with a ball-and-stick model. The basic residues located at the DNA-binding site of CMD1 are shown with ball-and-stick models. The basic residues located in adjacent to the DNA-binding site of CMD1 that might have potential interactions with the DNA are also shown with ball-and-stick models and circled with black dashed line. **f** Effect of mutations of the key residues involved in the DNA binding on the enzymatic activity of CMD1 and the binding affinities of CMD1 toward dsDNA. ND not detected. Data are presented as mean values ± SEM. $n = 3$ independent replicates.

CMD1 without the MBP tag ($2.01 \pm 0.11\,\mu M$; Supplementary Fig. 6d). Moreover, although the two types of CMD1 exhibit comparable activities towards the 14-nt DNA substrate, the MBP-fused CMD1 is less efficient in catalyzing the 0.5-kb DNA substrate than CMD1 without the MBP tag (Supplementary Fig. 6e). These results may explain why in the structure of CMD1 in complex with the 14-nt DNA, a large portion of the DNA is defined with high-quality electron density, whereas in the structures of CMD1 in complexes with DNAs with a shorter or longer overhang, only a small portion of the DNA is defined.

## Discussion

The TET homologue CMD1 in the green alga *C. reinhardtii* utilizes VC as the co-substrate to catalyze the conjugation of a glyceryl moiety to the $C^5$-methyl group of 5mC, forming a new DNA base modification named as 5gmC[26]. The 5gmC modification exists in the genomic DNA of the wild-type green alga, and CMD1 is responsible for maintaining the 5gmC level in vivo. Functional studies demonstrated that 5gmC is a potential epigenetic mark and plays a vital role in the regulation of photosynthesis[26]. In this work, we carried out extensive structural and biochemical studies of CMD1, which not only reveal the molecular basis for how CMD1 recognizes the DNA substrate and utilizes VC as the co-substrate, but also provide mechanistic insight into the catalytic reaction of the 5gmC DNA modification by CMD1.

Comparison of the overall structures of the DNA-bound CMD1, *Homo sapiens* TET2 (HsTET2, PDB code: 4NM6)[38] and *Naegleria gruberi* TET1 (NgTET1, PDB code: 4LT5)[41] shows that the DSBH core of CMD1 is similar to that of HsTET2 (RMSD of 3.3 Å for 214 Cα atoms) and NgTET1 (RMSD of 3.0 Å for 196 Cα atoms) despite the fact that the flanking structure elements are quite different and could not be aligned very well (Fig. 4a). In all of these enzymes, the DNA substrate always binds to a positively charged surface cleft, and the protein interacts mainly with the phosphate backbone of the DNA substrate via extensive hydrophilic interactions (Fig. 3a; refs. [38,41] and this study).

A detailed structural comparison reveals that the active site of CMD1 has some commonalities, as well as some differences with that of HsTET2 and NgTET1. Particularly, despite the low overall sequence identity (Supplementary Fig. 7), the conserved HXD/E…H motif of CMD1, HsTET2, and NgTET1 could be structurally aligned very well (Fig. 4b, c). Nevertheless, although the residues composing the active sites could be structurally aligned well, the chemical properties of some residues are different, and thus there are substantial differences in the binding and recognition mode of the co-substrate. Specifically, in the structures of CMD1 in complexes with VC and with both DNA and VC, the VC coordinates the $Fe^{2+}$ in a monodentate manner via its C-2

hydroxyl group (Supplementary Fig. 5a and Fig. 2b). This is significantly different from the binding mode of the 2-OG analog NOG, which coordinates the $Fe^{2+}$ in a bidentate manner via its C-1 carboxyl and C-2 ketone in the HsTET2 and NgTET1 structures[40,41] (Fig. 4b). In addition, in the DNA-bound HsTET2 and NgTET1 structures, the NOG is stabilized by two salt bridges and a number of direct hydrogen-bonding interactions (Supplementary Fig. 8a, b). However, in the CMD1–5mC–DNA–VC complex, the VC is stabilized by a few direct hydrogen-bonding interactions and several indirect hydrogen-bonding interactions via water molecules (Supplementary Fig. 8c). These results suggest that VC makes less hydrophilic interactions with CMD1 than NOG with HsTET2 and NgTET1, which might explain why CMD1 has weak binding affinity for VC.

In the HsTET2 and NgTET1 structures, the C1-carboxyl and C5-carboxyl of NOG are each stabilized by an Arg residue (Arg1261 and Arg1896 of HsTET2, and Arg224 and Arg289 of NgTET1, respectively), and these two Arg residues are strictly conserved and are characteristic basic residues responsible for the 2-OG binding in the $Fe^{2+}$/2-OG-dependent dioxygenases[24,39]. Although CMD1 contains two Arg residues at the active site (Arg244 and Arg410) and the structure-based sequence alignment shows that these Arg residues can be aligned with those of HsTET2 and NgTET1 (Supplementary Fig. 7), only the side chain of Arg244 makes an indirect hydrogen-bonding interaction with the VC, while the side chain of Arg410 points away from the active site and forms several hydrophilic interactions with the surrounding residues and thus makes no interaction with the VC (Fig. 4b). In addition, the side chain of Trp358 occupies the spatial position of the side chain of Arg1896 of HsTET2 or Arg289 of NgTET1, and has potential steric conflict with the C5-carboxyl of NOG (Fig. 4b). Thus, due to the lack of a basic residue to stabilize the C5-carboxyl of 2-OG and the potential steric hindrance of the side chain of Trp358, 2-OG would not be able to bind properly at the active site of CMD1. This may explain why CMD1 could not use 2-OG as co-substrate. On the other hand, the shape of the co-substrate binding pocket of CMD1 is substantially different from that of the TET proteins (Supplementary Fig. 8d), and VC would have potential steric conflicts with some residues constituting the co-substrate binding pocket of the TET proteins (Fig. 4b). This may explain why the TET proteins could not use VC as co-substrate. Taken together, our structural and biochemical data show that although CMD1 is a TET homologue belonging to the $Fe^{2+}$/2-OG-dependent dioxygenase family, the key residues of CMD1 involved in the VC binding are quite different from those of HsTET2 and NgTET1 involved in the NOG (or 2-OG) binding, and the binding and recognition mode of VC by CMD1 is thus substantially different from that of NOG

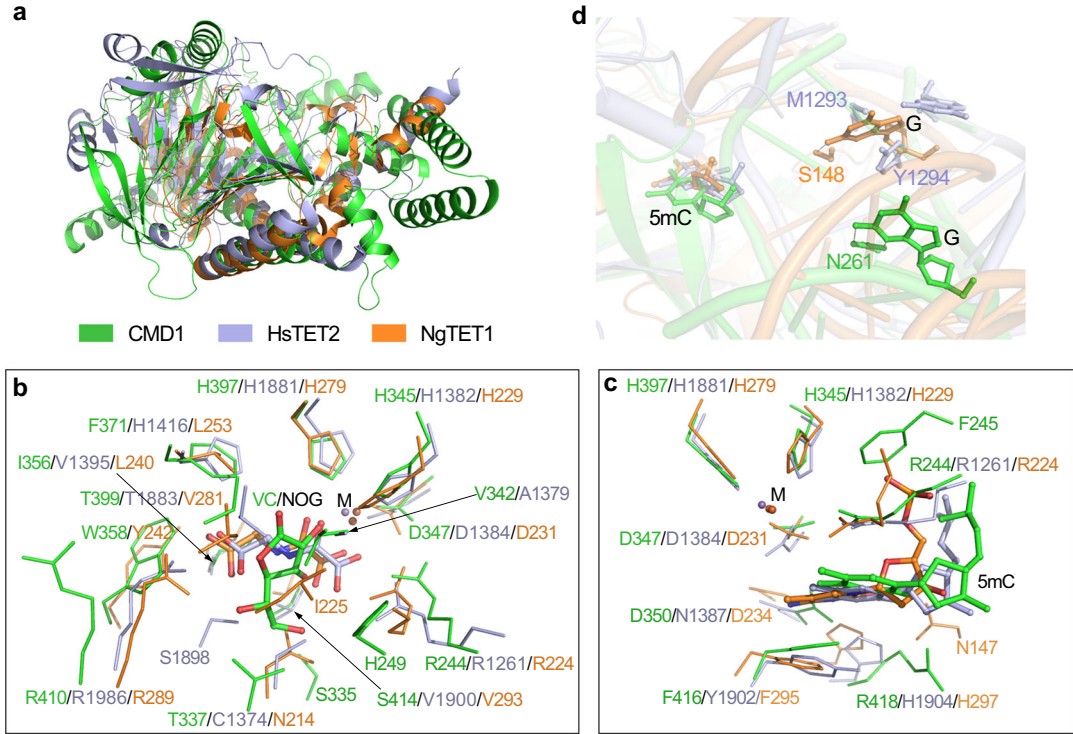

**Fig. 4 Structural comparison of CMD1 with TET proteins. a** Comparison of the overall structures of CMD1, HsTET2 (PDB code, 4NM6) and NgTET1 (PDB code, 4LT5). CMD1, HsTET2, and NgTET1 are colored in green, light-blue, and orange, respectively. **b, c** Comparison of the co-substrate binding sites (**b**) and the substrate binding sites (**c**) of CMD1, HsTET2, and NgTET1. The co-substrate and residues composing the active site are shown with ball-and-stick models. **d** Comparison of the 5mC base flipping in the structures of CMD1, HsTET2, and NgTET1.

by HsTET2 and NgTET1. These differences could explain why CMD1 could not use 2-OG and why the TET proteins could not use VC as co-substrate.

On the other hand, structural comparison shows that the binding and recognition mode of 5mC by CMD1 shares some commonalities with that by HsTET2 and NgTET1 (Fig. 4c). Structure-based sequence alignment shows that the residues involved in the 5mC binding can be aligned well among CMD1, HsTET2, and NgTET1, and the aligned residues share similar chemical properties (Fig. 4c and Supplementary Fig. 7). Specifically, the N4 atom of the 5mC base is stabilized by an Asp or Asn (Asp350 of CMD1, Asn1387 of HsTET2, and Asp234 of NgTET1) via a hydrogen bond. The N3 atom of the 5mC base is recognized by a His (His1904 of HsTET2 and His297 of NgTET1) with a hydrogen bond, and the O2 atom of the 5mC base is stabilized by an Arg or Asn (Arg418 of CMD1 and Asn147 of NgTET1) via a hydrogen bond. In addition, the pyrimidine ring of the 5mC base makes a π–π stacking interaction with the side chain of a Phe or Tyr (Phe416 of CMD1, Tyr1902 of HsTET2, and Phe295 of NgTET1) from the bottom. Nonetheless, the pyrimidine ring of the 5mC base also makes a cation–π stacking interaction with the side chain of Arg244 on the top in the CMD1–DNA complexes, and a partial cation–π stacking interaction with the side chain of Arg1261 and Arg224 in the HsTET2–DNA and NgTET1–DNA complexes, respectively (Fig. 4c). These results suggest that the TET and TET-like proteins utilize a conserved manner to bind and recognize 5mC.

In the DNA-bound CMD1, HsTET2, and NgTET1 complexes, the 5mC base is flipped out of the dsDNA and inserted into the active site (Fig. 4d). In the HsTET2–DNA complex, the orphaned guanine base is also flipped out of the dsDNA and the side chains of Met1293 and Tyr1294 occupy the position of the guanine

base[38] (Fig. 4d). It was suggested that the side chains of Met1293 and Tyr1294 might push the guanine base out of the dsDNA, which would disrupt the hydrogen-bonding interactions of the 5mC:G base pair, and thus facilitate the flipping of the 5mC base[38]. However, in the CMD1–DNA and NgTET1–DNA complexes, the orphaned guanine base maintains an intra-helical conformation and forms a direct hydrogen bond with the side chain of Asn261 of CMD1 or Ser148 of NgTET1 (Fig. 4d). In addition, in the CMD1–DNA complexes, there is no large side chain residue located in the vicinity of the 5mC/C:G base pair. Our biochemical data also shows that mutation of Asn261 to Ala leads to about fivefold reduction of the CMD1 activity (Fig. 3f), suggesting that the hydrophilic interaction between the side chain of Asn261 and the orphaned guanine base plays an important role in the stabilization of the orphaned guanine base and the catalytic reaction. It is possible that the flipping of the 5mC/C base might take place during the transient breakage of hydrogen bonds, namely hydrogen bond breathing in DNA duplex[42]. When the 5mC/C base is flipped out of dsDNA and inserted into the active site of CMD1, it would be stabilized by Arg244 and Phe245, and meanwhile the orphaned guanine base would be stabilized by the side chain of Asn261. It is also possible that the interaction between the side chain of Asn261 and the guanine base might weaken the hydrogen bonds of the 5mC/C:G base pair, and thus facilitate the base flipping process.

In the well-established catalytic mechanism of the $Fe^{2+}$/2-OG-dependent dioxygenases[23–25], the $O_2$ molecule is first activated by the metal ion and then the activated oxygen attacks the C2-oxo group of co-substrate 2-OG, leading to the cleavage of the C1–C2 bond of 2-OG to produce the co-products succinate and $CO_2$ and the formation of a ferryl-oxo ($Fe^{IV}=O$ or $Fe^{III}–O^-$) species. Subsequently, the ferryl-oxo species abstracts a hydrogen atom

from the substrate, leading to the formation of a substrate radical and a Fe(III)-hydroxide species. Finally, the substrate radical and the hydroxide join together to form a hydroxylated product. In the CMD1–5mC–DNA–VC structure, a water molecule (Wat3) mediates the hydrophilic interactions between the side chain of Ser414 and the C3-hydroxyl group of VC (Fig. 2d). It is possible that Wat3 might occupy the position of the $O_2$ molecule in the catalytic reaction. In addition, Arg244 makes interactions with the conserved Asp347, the VC, and the 5mC base (Fig. 2d). Our biochemical data shows that the R244A mutant is unable to catalyze the formation of 5gmC (Fig. 2c). These results indicate that Arg244 plays a vital role not only in the stabilization of the 5mC base and the binding of VC, but also in the catalytic reaction.

The previous biochemical studies have shown the metabolic destiny of VC in the reaction catalyzed by CMD1 (ref. [26]). Briefly, the $CO_2$ is derived from the C1 atom of VC; the glyoxylic acid is derived from the C2 and C3 atoms of VC; and the glycerol moiety added to the 5mC is derived from the C4–C6 atoms of VC. These results suggest that the C1–C2 and C3–C4 bonds of VC are cleaved and the 5mC methyl radical attacks the C4 atom of VC to form the 5gmC product in the catalytic reaction. However, in the structure of CMD1–5mC–DNA–VC complex, VC is present in the lactone form and the catalytic reaction has not taken place, which is consistent with the biochemical data showing that CMD1 is inactive in the crystals and crystallization drops. Our previous biochemical studies showed that the CMD1 activity could not be supported by analogs or derivatives of VC[26], and our current biochemical studies also showed that the ring-opened form of VC (2-keto-L-gulonate) and the decarboxylated intermediate (L-xylonic acid) could not support the activity of CMD1 either, indicating that the glycerol transfer reaction catalyzed by CMD1 could only take place with the lactone form of VC. Thus, the CMD1–5mC–DNA–VC complex is very likely to represent the initial state of the catalytic reaction (Supplementary Fig. 9, step 1). During the catalytic reaction, the lactone form of VC would open the ring at specific condition(s) and the ring-opened form of VC might maintain the mono-coordination with the metal ion or undergo some conformational changes to coordinate the metal ion in a bidentate manner similar to 2-OG (Supplementary Fig. 9, step 2). Then, the ring-opened form of VC is decarboxylated to generate L-xylonic acid (or other intermediate) upon the activation of $O_2$ by the metal ion using a similar mechanism as the other $Fe^{2+}$/2-OG-dependent dioxygenases (Supplementary Fig. 9, step 3). In the CMD1–5mC–DNA–VC complex, the C4 atom of VC is positioned from the C5-methyl group of 5mC by 5.7 Å, and the C4–C6 moiety of VC is positioned even more distantly from the C5-methyl group of 5mC. Thus, it is very likely that the C4–C6 moiety of the ring-opened form or intermediate of VC would change its conformation to swing toward the C5-methyl group of 5mC (Supplementary Fig. 9, steps 4 and 5), permitting the transfer of the C4–C6 moiety of VC to the C5-methyl group of 5mC. In this process, the ring-opened form or other intermediate of VC might coordinate the metal ion in a bidentate mode similar to 2-OG. A modeling study shows that when L-xylonic acid assumes a bidentate coordination manner similar to 2-OG and is stabilized by the surrounding residues, the distance between the C4 atom of L-xylonic acid and the C5-methyl group of 5mC would be within 4 Å (Supplementary Fig. 8e), and the C4 atom could be attacked by the 5mC radical to generate 5gmC and glyoxylic acid (Supplementary Fig. 9, step 6). Nevertheless, based on the available biochemical and structural data, we do not know exactly at what stage of the catalytic reaction the ring of VC is hydrolytically opened and what kind of ring-opened intermediate would be generated and employed in the catalysis. Hence, the exact and detailed molecular mechanism of the catalytic reaction by CMD1 could not be deduced and warrants further biochemical and structural studies.

## Methods

**Cloning, expression, and purification of CMD1.** The N-terminal $His_6$-SUMO-tagged full-length CMD1 (residues 1–532) was used in the enzymatic activity assay and BLI assay. For expression of CMD1 used in the crystallization, the gene encoding the full-length CMD1 was subcloned into the V28E4 vector[43], which attaches a MBP tag and a $His_6$ tag at the N-terminus and C-terminus of CMD1, respectively. The primers used in subcloning are summarized in Supplementary Table 6. The constructed plasmids were transformed into *E. coli* BL21(DE3) Condon Plus strain. The transformed bacterial cells were grown in LB medium at 37 °C to $OD_{600}$ of 0.6–0.8, and then induced with 0.2 mM IPTG at 16 °C overnight. The cells were collected, resuspended, and lysed on ice by high-pressure crushing in buffer A (287 mM NaCl, 2.7 mM KCl, 1 mM $Na_2HPO_4$, and 2 mM $KH_2PO_4$) or buffer B (20 mM HEPES, pH 7.5, and 300 mM NaCl) supplemented with 2 mM β-mercaptoethanol and 1 mM phenylmethylsulfonyl fluoride. The cell debris was precipitated by centrifugation and the supernatant was collected for protein purification.

The $His_6$-SUMO–CMD1 protein was firstly purified by affinity chromatography using a Ni-NTA column (Qiagen) pre-equilibrated with buffer A, and then the purified protein was incubated overnight with ULP1 protease at 4 °C to cleave the $His_6$-SUMO tag. The CMD1 protein was further purified with gel filtration using a Superdex G200 10/300 column (GE Healthcare) pre-equilibrated with buffer C (20 mM HEPES, pH 7.0, and 100 mM NaCl). The MBP–CMD1–$His_6$ protein was purified by affinity chromatography using a Ni-NTA column with buffer B supplemented with 20 mM imidazole and 200 mM imidazole serving as the washing buffer and elution buffer, respectively. The MBP–CMD1–$His_6$ protein was further purified with gel filtration using a Superdex G200 10/300 column pre-equilibrated with buffer D (20 mM HEPES, pH 7.5, 150 mM NaCl, and 1 mM dithiothreitol). The Se-Met derivative MBP–CMD1–$His_6$ protein was expressed and purified the same as the native protein except that the bacterial cells were grown in M9 medium. Constructs of the CMD1 mutants containing point mutations were generated using the QuikChange Site-Directed Mutagenisis kit (Stratagene) and verified by sequencing. Expression and purification of the mutants were the same as the wild-type proteins. The purified proteins were of high purity (>95%) as analyzed by SDS–PAGE (10%).

**Crystallization, data collection, and structure determination.** Prior to crystallization, the purified MBP-tagged CMD1 was concentrated to ~30 mg/ml in buffer D and then incubated with $(NH_4)_2Fe(SO_4)_2$ at a molar ratio of 1:2. Crystallization was carried out using the hanging drop vapor diffusion method at 16 °C. Crystals of the Se-Met derivative CMD1 were grown in drops consisting of equal volumes of the protein solution and the reservoir solution [0.1 M sodium citrate, pH 5.5, and 16% (w/v) PEG 8000]. Crystals of CMD1 in apo form were grown in drops consisting of the protein solution and the reservoir solution [2% (v/v) tacsimate, pH 5.0, 0.1 M sodium citrate, pH 5.6, and 16% (w/v) PEG 3350]. Crystals of CMD1 in complex with DNA were grown in drops consisting of the protein solution pre-incubated with synthesized DNA substrates at a molar ratio of 1:1.2 and the reservoir solution [15% (v/v) 2-propanol, 0.1 M sodium citrate, pH 5.0, and 10% (w/v) PEG 10,000]. To obtain VC-bound CMD1 and CMD1–5mC–DNA complexes, the crystals of the apo CMD1 or the CMD1–5mC–DNA complex were soaked in the reservoir solution supplemented with 400 mM VC (sodium L-ascorbate) for 5 min, and then transferred to the cryoprotectant consisting of the reservoir solution and 30% ethylene glycol followed by flash-cooling into liquid $N_2$. Diffraction data were collected at BL17U1 of Shanghai Synchrotron Radiation Facility or BL18U and BL19U1 of National Facility for Protein Science Shanghai, using Blu-Ice as the control and data collection software, and processed with HKL2000 (ref. [44]). The statistics of the diffraction data are summarized in Table 1.

The crystal structure of the Se-Met derivative CMD1 was solved using the SAD method as implemented in Phenix[45]. All of the other structures were solved by the MR method using the Se-Met derivative CMD1 or the apo CMD1 structure as the search model. Model building was performed with Coot[46], and structure refinement was carried out using Refmac5 (ref. [47]) and Phenix[45]. Structural analysis was carried out using programs in CCP4 (ref. [48]). The structure figures were generated using Pymol (www.pymol.org)[49]. The statistics of the structure refinement and the quality of the final structure models are also summarized in Table 1.

**Preparation of DNA substrates.** For co-crystallization of CMD1 and DNA, a 14-nt ssDNA with a sequence of 5′-CC(mC)GCGCGGGATGT-3′ was synthesized and annealed to obtain a 10-bp palindromic dsDNA plus a 4-nt 3′-overhang on both the substrate strand and the non-substrate strand. All of the DNA oligos used in the enzymatic activity assay and BLI assay were synthesized and annealed properly, and the sequences of the oligos are summarized in Supplementary Tables 1 and 2. The 0.5-kb 5mCpG- and CpG-containing DNA substrates used in the enzymatic activity assay were amplified by PCR from a randomly selected portion of *C. reinhardtii* genomic DNA using 5-methyl-dCTP and dCTP, respectively. The forward primer (DNA-Sub-F) and the reverse primer (DNA-Sub-R) are summarized in Supplementary Table 6. The 0.5-kb 5mCpG- and CpG-containing DNA substrates contain ~25.5% T base.

**Bio-layer interferometry assay.** The binding affinities of the full-length CMD1 with different DNA substrates were analyzed by the BLI assay using Octet Red 96

(ForteBio). The synthesized 5'-biotin labeled DNAs (25 nM) were immobilized on Streptavidin biosensor (SAs, ForteBio) and the typical immobilization levels were ~0.2–0.3 nm. The ligands-loaded SAs were then incubated with decreasing amount of the wild-type or mutant CMD1 protein in the buffer containing 20 mM Tris-HCl (pH 8.0), 50 mM NaCl, 0.1% Tween-20, and 0.5% BSA. All experiments were performed in duplicates and carried out in solid-black-96-well plates containing 200 μl of solution in each well with an agitation speed of 1000 r.p.m. at 25 °C. The $K_D$ values were obtained by global fitting of the binding curves with the 1:1 binding model using the Octet data analysis software 9.0.

**In vitro enzymatic activity assay.** The enzymatic activity assay of the full-length and MBP-fused CMD1 was performed at 37 °C using a modified method described previously[26]. The reaction mixture of 100 μl consisted of 7 μM wild-type or mutant CMD1 protein, 1 mM L-ascorbic acid, and 8 nM biotinylated 0.5-kb 5mC/C-DNA or 2 μM biotinylated 90-bp 5mC-CpX DNAs or 2 μM biotinylated 14-nt 5mC/C-DNA in the reaction buffer [50 mM HEPES, pH 7.0, 50 mM NaCl, 0.1 mM Fe (NH$_4$)$_2$(SO$_4$)$_2$, and 2 mM Tris(2-carboxyethyl)phosphine]. After 2 h reaction, the mixture was treated with proteinase K (Lifefeng) at 55 °C for 1 h, and the biotinylated DNA was purified by Streptavidin Sepharose beads (GE Healthcare) according to the manufacturer's instructions. In the analysis of the effects of different buffers with different pH values on the activity of CMD1, the HEPES buffer (pH 7.0) in the reaction mixture was substituted with other buffers with different pH values, including sodium citrate (pH 5.0 and 6.0), sodium cacodylate (pH 5.0, 6.0, and 7.0), MES (pH 6.0 and 7.0), BIS-TRIS (pH 6.0 and 7.0), and HEPES (pH 8.0).

The purified DNA was digested into nucleoside hydrolysates by nuclease P1 (Sigma) at 55 °C for 1 h in the buffer containing 20 mM NaAc (pH 5.3) and 0.2 mM ZnSO$_4$, and then dephosphorylated with calf intestinal alkaline phosphatase (Takara) at 37 °C for 1 h. The digested mixture was centrifuged and then subjected to LC–MS analysis. The supernatant was analyzed using the multi-reaction monitoring (MRM) mode of LC/MS QQQ (Agilent 1260/6495B tandem mass spectrum). Metabolite standards were used to identify the metabolites of the catalytic reaction. A Hypersil GOLD aQ column (1.9 μm particle size, 150 mm × 2.1 mm, Thermo Scientific) was used for LC separation using gradient elution with water supplemented with 0.1% formate as solvent A and acetontrinle supplemented with 0.1% formate as solvent B. The gradient program was as follows: 0–5 min 100% A, 5–10 min 100–92% A, 10–12 min 92–0% A, 12–15 min 100% B, and 15–15.1 min 100% A. The flow rate was 0.3 ml/min and the column temperature was 25 °C. An AJS ESI ion source in positive ion mode was used for detection. Nitrogen generator (PEAK Shanghai) was used for solvent removal and atomization, and high purity nitrogen as colliding gas. sheath gas temp was 250 °C, sheath gas flow 11 l/min, gas temp 300 °C, and gas flow 14 l/min. Capillary voltage was 4000 V, nebulizing gas 20 p.s.i., nozzle voltage 1500 V, and collison energy 5 eV. Selected ions for nucleosides were as follows: MRM transitions 228.1 → 112.1, 242.1 → 126.1, 258.1 → 142.1, 272.1 → 156.1, 332.1 → 150.1, and 243.1 → 127.1 for the detection of C, 5mC, 5hmC, 5caC, 5gmC, and T, respectively. The expected MRM transition for 5gT or 5gmU is at 333.1 → 217.1. The quantities of the produced 5gmC and the remaining 5mC were semi-quantified based on integrations of the 5gmC peak and the 5mC peak. The enzymatic activity of CMD1 is represented by the ratio of the 5gmC over the total of the 5gmC and the remaining 5mC. All experiments were performed in triplicates and the error bars represent the standard error of the mean. The graphs were generated with GraphPad Prism.

**Reporting summary**. Further information on experimental design is available in the Nature Research Reporting Summary linked to this paper.

## Data availability
The crystal structures of CMD1 in apo form and in complexes with VC, with 5mC–DNA, with DNA, and with both 5mC–DNA and VC have been deposited with the Protein Data Bank under accession codes 7CY4, 7CY5, 7CY6, 7CY7, and 7CY8, respectively. Other data are available from the corresponding author upon reasonable request. Source data are provided with this paper.

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

## Acknowledgements

We thank the staff members at BL17U1 of Shanghai Synchrotron Radiation Facility of China and BL18U and BL19U1 of National Facility for Protein Science Shanghai for technical supports in diffraction data collection, and the staff members at the Core Facility of Molecular Biology of our institute for technical supports in biochemical studies. We are grateful to Dr. Jin-Qiu Zhou, Dr. Suwen Zhao, and other members of our group for valuable discussion. This work was supported by grants from the National Key Research and Development Program of China (2020YFA0509000) and the Strategic Priority Research Program of Chinese Academy of Sciences (XDB37030305). W.L. acknowledges the support of the SA-SIBS scholarship program.

## Author contributions

J.D. conceived the study. W.L. and M.S. carried out the protein preparation, DNA binding assays, and crystallization experiments. W.L., M.S., and T.Z. collected the diffraction data. W.L. and T.Z. carried out the structure determination and refinement. M.S. carried out the mutagenesis and enzymatic activity assays. Y.S. and X.-J.Z. participated in the biochemical studies. G.-L.X. participated in the data analyses and discussion. W.L., T.Z., and J.D. designed the experiments, performed the data analyses, and wrote the manuscript with contributions from all of the other authors.

## Competing interests

The authors declare no competing interests.
