## [Peer Review File · Nature Communications]

REVIEWER COMMENTS

Reviewer #1 (Remarks to the Author):

A TET homologue, named as CMD1, in a green alga *Chlamydomonas reinhardtii*, was found to exhibit a novel enzymatic activity to catalyze the conjugation of a glyceryl moiety to the C5-methyl group of 5mC via a direct carbon-carbon bond, leading to a new DNA base modification named as C5-glyceryl-methylcytosine (5gmC).

The authors employ structural and biochemical means to reveal the underlying molecular mechanism for the VC-derived 5gmC DNA modification by CMD1. The work is extensive and conclusive.

Some concerns:

1. The major novelty of this study is the structural basis of CMD1-mediated catalysis on VC but not 2OG. The author should make clearer discussion and conclusion why CMD1 could not use 2OG and why other Fe/2OG-dependent enzymes could not use VC as co-substrate. It would be much more important if some mutations of CMD1 allow the enzyme to use 2OG for reaction (or TET mutation to use VC). This would be very challenging and I don't expect this. But this would not only significantly highlight the value of this work, but also provide fundamental structural basis for the whole field of Fe/2OG-dependent enzymes.

2. The electron density around VC is relatively weak. I guess that authors have tried very hard to improve the resolution. Although it is reasonable to put a VC molecule into this density, discussion of intermolecular interactions between VC and CMD1 should be careful. There might be some errors under the density map at this resolution. I would suggest the authors may tone down their description about the interactions (such as hydrogen bonds).

3. The CMD1 binds DNA substrate via interaction with the base and the phospho-ribose backbone. Therefore, the phospho-ribose interaction with DNA substrate might dominate the binding affinity. To check the influence of DNA sequences on enzyme activity, enzymatic assay would be more informative. The authors might perform enzymatic assay to show whether 5mC in different context could affect enzyme activity. For example, HsTET prefers 5mC in CG context. The CMD1 preference could be addressed in the manuscript.

Reviewer #2 (Remarks to the Author):

CMD1 is a TET related enzyme that catalyzes a partly similar and partly distinct reaction, that differs from the TET catalyzed reaction in the choice of one of the co-substrates (vitamin C instead of 2OG) and in the reaction products. Mechanistically, the reactions catalyzed by CMD1 and TETs are similar in the early stages, but very different at the later stages. Due to the novel enzymology, structures of CMD1 are of great interest to the community.

Here, the authors, among them the discoverers of CMD1, present a very complete set of structures of CMD1 alone, in complex with VC, in complex with dsDNA (methylated and non-methylated), and in complex with methylated dsDNA and VC. The structures are at moderate resolution. As is often the case, the mechanistically most important complex (with 5mC dsDNA and VC) is at lower resolution than the others. Although the resolution of all complexes is not ideal for a detailed mechanistic interpretation, the structures represent a remarkable achievement. Standard crystallographic quality indicators suggest that data collection has been carefully done, and overall refinement quality indicators are also fine (but see comment below on steric conflicts of implicit hydrogen atoms). Functional assays with enzyme variants confirm the role of important residues, further validating the essential correctness of the structures.

Although the authors have extremely interesting data in their hands, they could make yet more of it. Many aspects of the structure that are important for a precise mechanistic understanding of the CMD1 reaction, such as the accuracy of ligand binding poses, and the identity of the active site metal cation, deserve more attention (and in some cases, simply presentation in the main text, rather than in the Suppl). Most importantly, the authors should aim to clearly place their CMD1-

5mC dsDNA-VC complex in the catalytic scheme.

In some places, the text could benefit from (minor) polishing by a native speaker.

Major comments:

=====

- Please be precise about the cation in the active site. Ideally, it should be asserted by anomalous diffraction, looking at the anomalous difference density at wavelengths above and below the absorption edge of candidate metal ions. For Fe²⁺, this is not easy, because the relevant absorption edge is at 1.74 Å, and may be hard to reach on many synchrotron sources (leaving aside problems with air absorption). If an absolutely certain metal ion identification from anomalous densities was possible, this would greatly add to the paper. However, it should not be a requirement, and I would also accept that the authors simply state more clearly why they believe that their preparation really contains the physiologically relevant iron (doubts arise because of the lack of activity in the crystal).

- If the metal ion in the active site is indeed Fe²⁺ as suggested in the text, and if crystals were grown without special protections to maintain anaerobic conditions (as Materials and Methods indicates) why does the density suggest a substrate complex? Are the crystals capable to catalyze the reaction, or do they trap a catalytically incompetent conformation? Activity of crystals (or protein in the crystallization drops) could be tested by mass spectrometry of bound DNA, possibly after digestion to single 2'-deoxynucleotides, and alkaline phosphatase treatment, to improve signal. If this is not possible because crystallization drops are no longer available, the authors could determine (or cite, if it's already available) a pH profile of the reaction to test whether their enzyme should be active under crystallization conditions. In principle, the acidic conditions in the crystallization drop (pH 5.0, see p. 24, line 539) should be protective for the Fe²⁺ resting state (CMD1 inactivation would be expected in basic, but not acidic conditions). If activity is seen in the crystal, or at pH 5.0, why does one not get the electron density for a product complex? If activity is not seen at pH 5.0, can this be explained in structural terms? In other words, which protonations would prevent catalysis? And what caveats does this create for the interpretation?

- The reader needs to see the strength of your evidence for the binding mode of VC, C and 5mC. Please show composite omit densities in the main text figures. Suppl. Fig. 4, panel B is perfect in this respect for VC, a sub-region of Suppl. Fig. 4 panels C and D would do for the bases. For the complex with 5mC dsDNA and VC, it is crucial to see the density evidence that the lactone form of VC is present in this complex, and that glyceryl transfer to 5mC has not taken place (unless it has already been shown biochemically that the reaction does not occur in crystallization conditions, see above).

- Like the TET reaction, the CMD1 reaction involves an oxidative decarboxylation. Comparison with the TET reaction suggests that VC is hydrolytically ring opened, and that the ring-opened form (an alpha-keto-acid) is then oxidatively decarboxylated. However, the text and Fig. 2B suggest that even in the complex with 5mC dsDNA and VC, VC is still present in the lactone form. Could this be misleading and related to hindered catalysis?

- The Discussion on catalytic mechanism is the most interesting part of the entire work. This core part of the paper is hard to understand without a scheme that shows the reader the numbering scheme, ideally for lactone and ring-opened forms of VC. Could this be added as an inset to one of the figures? Alternatively, can you introduce the numbering scheme with the figure that shows VC?

- I concur with the authors that the CMD1 reaction cleaves the C1-C2 and C3-C4 bonds in vitamin C. However, I always thought that the only plausible reaction mechanism requires an attack of the 5mC methyl radical on the C4. Attack on C3 would create different reaction product. The relevant distance should therefore be the distance from the methyl group to C4, not to C3. Yet, the authors discuss both distances as if they were equally relevant for catalysis. If indeed they think so, they should explain the mechanistic rationale for why the methyl-C3 distance should matter. Otherwise, they should focus on the methyl-C4 distance.

- The methyl-C4 distance (and the geometry of radical attack) could be the highlight of the manuscript. Unfortunately, it is not clear whether this distance can be meaningfully deduced from the current structures. The authors suggest, and I completely agree, that VC decarboxylation must precede, not follow methyl radical attack on decarboxylated VC (the radical generator Fe(IV)oxo is created with participation of one oxygen atom from molecular oxygen, the other is involved in the decarboxylation). As the methyl-C4 distance is so central to the chemistry, the information on this distance should remain in the manuscript. It is important enough that the limitations of the current distance determination (possible changes of VC pose after ring opening) should be discussed (or at least mentioned).

- Mechanistically, the reaction CMD1 reaction requires a pyrimidine 5-methyl group. The enzyme therefore does not need to distinguish between 5mC and C to selectively transfer glyceryl to 5mC. Hence, the similar affinity of CMD1 to 5mC and C containing dsDNA is perhaps not so surprising. In contrast to C, T has a 5-methyl group placed on the pyrimidine scaffold. Hence, in rare cases the CMD1 reaction could happen with T instead of 5mC. For TETs, the Carell group has shown that similar errors indeed occur at low frequency. Does this happen with CMD1? If I interpret Fig. 2D correctly, the preference for 5mC over T is due to a favorable hydrogen bond from the N4 of 5mC to Asp350. The authors have the Asp350Asn variant (see Fig. 2C). If the wild-type protein does not do it, does this variant transfer glyceryl to T, creating a novel DNA base? While this may be an easy experiment to do given that the authors have the enzymes in hand, it is not essential experiment for the main message of the manuscript and therefore should not be required for publication.

Technical comments:

=====

The PDB validation reports contain long lists of (moderate) steric conflicts, almost all of which are caused by implicit hydrogen atoms. At this resolution, this is not a major problem, because hydrogen positions cannot be relied on anyway. Nevertheless, the list of conflicts is oddly long. The authors may want to check their hydrogen model used in refinement, to clarify whether this is a PDB issue, or whether there are genuine conflicts that could be avoided.

Minor editorial comments:

=====

Throughout: the authors have not decided between the "mC" and "5mC" nomenclature for 5-methylcytosine. Moreover, in some cases, they add the residue numbering in the structure that converts mC to mC3, according to the sequence position. If possible, this inconsistency should be avoided.

p.3, line 48: "the epigenetics field has attracted increasing interests to the discovery of additional DNA base modifications and their biological functions." Please reword.

p. 4, line 66: "consisting of a major and a minor beta-sheets" should be "consisting of a major and a minor beta-sheet" (no plural), but please double-check with a native speaker.

p.6, line 130: "of high AT level" should be "AT-rich", likewise for "GC level" later on.

p.9, paragraph: "Crystal structures of CMD1in complexes with VC, DNA, and both VC and DNA". Please state somewhere early in the paragraph whether the DNA contained a CpG or 5mCpG, in the substrate strand (now it takes inspection of Fig. 2 to see that a 5mC was in the substrate strand). Fig. 3C suggests that the non-substrate strand is non-methylated. Please introduce this early on in the text.

p. 10, line 207: "The crystals of CMD1 in complex with both DNA and VC were also obtained by soaking". This statement is easily misunderstood. I believe that you co-crystallized CMD1 with dsDNA, and then soaked in vitamin C. Please reword.

p. 18, line 393: "despite of the low sequence identity" should be "despite the low sequence identity"

Matthias Bochtler

Reviewer #1

Overall comments: A TET homologue, named as CMD1, in a green alga *Chlamydomonas reinhardtii*, was found to exhibit a novel enzymatic activity to catalyze the conjugation of a glyceryl moiety to the C5-methyl group of 5mC via a direct carbon-carbon bond, leading to a new DNA base modification named as C5-glyceryl-methylcytosine (5gmC).

The authors employ structural and biochemical means to reveal the underlying molecular mechanism for the VC-derived 5gmC DNA modification by CMD1. The work is extensive and conclusive.

Response: We thank the referee for the positive comments about our work.

Comments:

Comment 1. The major novelty of this study is the structural basis of CMD1-mediated catalysis on VC but not 2OG. The author should make clearer discussion and conclusion why CMD1 could not use 2OG and why other Fe/2OG-dependent enzymes could not use VC as co-substrate. It would be much more important if some mutations of CMD1 allow the enzyme to use 2OG for reaction (or TET mutation to use VC). This would be very challenging and I don't expect this. But this would not only significantly highlight the value of this work, but also provide fundamental structural basis for the whole field of Fe/2OG-dependent enzymes.

Response: We thank the reviewer for the constructive suggestion. As pointed out by the reviewer, it would be very challenge to identify some mutations of CMD1 which would allow the mutant(s) to use 2-OG for reaction (or mutations of TET to use VC). To attend the reviewer's comment, in the revision, we have made a more extended discussion about the differences in the binding of co-substrate between CMD1 and the TET proteins, which may explain why CMD1 could not use 2-OG and why other Fe²⁺/2-OG-dependent enzymes could not use VC as co-substrate as follows: "In addition, the side chain of Trp358 occupies the spatial position of the side chain of Arg1896 of HsTET2 or Arg289 of NgTET1 and has potential steric conflict with the C5-carboxyl of NOG (Fig. 4b). Thus, due to the lack of a basic residue to stabilize the C5-carboxyl of 2-OG and the potential steric hindrance of the side chain of Trp358, 2-OG would not be able to bind properly at the active site of CMD1. This may explain why CMD1 could not use 2-OG as co-substrate. On the other hand, the shape of the co-substrate binding pocket of CMD1 is substantially different from that of the TET proteins (Supplementary Fig. 8d), and VC would have potential steric conflicts with some residues constituting the co-substrate binding pocket of TET proteins (Fig. 4b). This may explain why TET proteins could not use VC as co-substrate." (Pages 20-21)

Comment 2. The electron density around VC is relatively weak. I guess that authors have tried very hard to improve the resolution. Although it is reasonable to put a VC molecule into this density, discussion of intermolecular interactions between VC and CMD1 should be careful. There might be some errors under the density map at this resolution. I would suggest the authors may tone down their description about the interactions (such as hydrogen bounds).

Response: We thank the referee for the constructive suggestion. In the revision, we have removed the description about the detailed hydrogen-bonding interactions between CMD1 and VC.

Comment 3. The CMD1 binds DNA substrate via interaction with the base and the phospho-ribose backbone. Therefore, the phospho-ribose interaction with DNA substrate might dominate the binding affinity. To check the influence of DNA sequences on enzyme activity, enzymatic assay would be more informative. The authors might perform enzymatic assay to show whether 5mC in different context could affect enzyme activity. For example, HsTET prefers 5mC in CG context. The CMD1 preference could be addressed in the manuscript.

Response: We thank the reviewer for the constructive suggestion. During the revision, we have carried out enzymatic activity assay using 90-bp dsDNAs containing six 5mCpX sites, where X is G, A, C, or T, to examine whether 5mC in different contexts would affect the activity of CMD1. The results show that CMD1 has a moderate substrate preference on 5mCpG-containing DNA. In the revised manuscript, we have added these results as follows: “It was reported previously that human TET2 exhibits a strong substrate preference for 5mCpG-containing DNA than 5mCpC- and 5mCpA-containing DNAs³⁸. To investigate whether 5mC in different contexts would affect the activity of CMD1, we carried out enzymatic activity assay using 90-bp dsDNAs containing six 5mCpX sites, where X is G, A, C, or T (**Supplementary Table 2**). The results show that CMD1 exhibits the highest activity on 5mCpG-containing DNA, relatively lower activity (about 50%) on 5mCpC- and 5mCpA-containing DNAs, and a much lower activity (about 15%) on 5mCpT-containing DNA (**Supplementary Fig. 1e**), suggesting that CMD1 has a moderate substrate preference for 5mCpG-containing DNA and could accommodate the substitution of guanine by adenine or cytosine to some extent.” (**Pages 7-8**)

Reviewer #2

Overall comments: CMD1 is a TET related enzyme that catalyzes a partly similar and partly distinct reaction, that differs from the TET catalyzed reaction in the choice of one of the co-substrates (vitamin C instead of 2OG) and in the reaction products. Mechanistically, the reactions catalyzed by CMD1 and TETs are similar in the early stages, but very different at the later stages. Due to the novel enzymology, structures of CMD1 are of great interest to the community.

Here, the authors, among them the discoverers of CMD1, present a very complete set of structures of CMD1 alone, in complex with VC, in complex with dsDNA (methylated and non-methylated), and in complex with methylated dsDNA and VC. The structures are at moderate resolution. As is often the case, the mechanistically most important complex (with 5mC dsDNA and VC) is at lower resolution than the others. Although the resolution of all complexes is not ideal for a detailed mechanistic interpretation, the structures represent a remarkable achievement. Standard crystallographic quality indicators suggest that data collection has been carefully done, and overall refinement quality indicators are also fine (but see comment below on steric conflicts of implicit hydrogen atoms). Functional assays with enzyme variants confirm the role of important residues, further validating the essential correctness of the structures.

Although the authors have extremely interesting data in their hands, they could make yet more of it. Many aspects of the structure that are important for a precise mechanistic understanding of the CMD1 reaction, such as the accuracy of ligand binding poses, and the identity of the active site metal cation, deserve more attention (and in some cases, simply presentation in the main text, rather than in the Suppl). Most importantly, the authors should aim to clearly place their CMD1-5mC dsDNA-VC complex in the catalytic scheme.

In some places, the text could benefit from (minor) polishing by a native speaker.

Response: We appreciate greatly the positive comments and constructive suggestions by the reviewer. In the revision, we have made extensive efforts to incorporate these suggestions (see our detailed responses below). In addition, we have asked some colleagues to help read and polish the manuscript. By doing so, the quality of the paper is greatly improved.

Major comments:

Comment 1. Please be precise about the cation in the active site. Ideally, it should be asserted by anomalous diffraction, looking at the anomalous difference density at wavelengths above and below the absorption edge of candidate metal ions. For Fe^{2+} , this is not easy, because the relevant absorption edge is at 1.74 Å, and may be hard to reach on many synchrotron sources (leaving aside problems with air absorption). If an absolutely certain metal ion identification from anomalous densities was possible, this would greatly add to the paper. However, it should not be a requirement, and I would also accept that the authors simply state more clearly why they believe that their preparation really contains the physiologically relevant iron (doubts arise because of the lack of activity in the crystal).

Response: We thank the reviewer for the constructive suggestion. As pointed out by the reviewer, we could not reach the relevant absorption edge of Fe^{2+} on the synchrotron source (SSRF) we used. The metal ion bound at the active site is provisionally assigned as Fe^{2+} owing to the presence of $(\text{NH}_4)_2\text{Fe}(\text{SO}_4)_2$ in the crystallization solution and the reasonable B-factor for the Fe^{2+} in the refinement. In addition, during the revision, we have performed the ICP-OES (inductively coupled plasma optical emission spectrometer) analysis to identify the types and abundances of metal ions in the protein solutions. The results show that Fe is the most abundant metal in the protein solution without or with supplementation of $(\text{NH}_4)_2\text{Fe}(\text{SO}_4)_2$ (Supplementary Table 4). Therefore, we believe that our protein preparation really contains the physiologically relevant iron and the bound metal ion is most likely an iron ion.

Our biochemical data show that CMD1 is inactive in the crystals because of the acidic condition of the crystallization solution (pH 5.0-5.5) but not the absence of the physiologically relevant iron ion (see our response to comment 2). In the revision, we have added the ICP-OES results as follows: “**There is a metal ion bound at the active site, which is interpreted as Fe^{2+} owing to the presence of Fe^{2+} in the protein solution and a reasonable B-factor for the Fe^{2+} in the refinement (Table 1). Indeed, the ICP-OES (inductively coupled plasma optical emission spectrometer) analysis shows that Fe is the most abundant metal in the protein solution without or with supplementation of $(\text{NH}_4)_2\text{Fe}(\text{SO}_4)_2$ (Supplementary Table 4), further supporting that the bound metal ion at the active site is an iron ion.**” (Page 9)

Comment 2. If the metal ion in the active site is indeed Fe²⁺ as suggested in the text, and if crystals were grown without special protections to maintain anaerobic conditions (as Materials and Methods indicates) why does the density suggest a substrate complex? Are the crystals capable to catalyze the reaction, or do they trap a catalytically incompetent conformation? Activity of crystals (or protein in the crystallization drops) could be tested by mass spectrometry of bound DNA, possibly after digestion to single 2'-deoxynucleotides, and alkaline phosphatase treatment, to improve signal. If this is not possible because crystallization drops are no longer available, the authors could determine (or cite, if it's already available) a pH profile of the reaction to test whether their enzyme should be active under crystallization conditions. In principle, the acidic conditions in the crystallization drop (pH 5.0, see p. 24, line 539) should be protective for the Fe²⁺ resting state (CMD1 inactivation would be expected in basic, but not acidic conditions). If activity is seen in the crystal, or at pH 5.0, why does one not get the electron density for a product complex? If activity is not seen at pH 5.0, can this be explained in structural terms? In other words, which protonations would prevent catalysis? And what caveats does this create for the interpretation?

Response: We thank the reviewer for the constructive suggestion. As pointed out by the reviewer, the original crystallization drops are no longer available, and thus we could not analyze whether the catalytic reaction took place in the crystals or the crystallization drops via MS analysis of the bound DNA. Nevertheless, in the revision, we followed the reviewer's suggestion and analyzed the effects of different buffers with different pH values on the activity of CMD1. The results show that CMD1 exhibits substantially differed activity in different buffers with different pH values. Among the conditions examined, CMD1 displays the highest activity in the HEPES buffer at pH 7.0 which was used in the enzymatic activity assay; however, it is inactive in the citrate buffers at pH 5.0-6.0 which were used in the crystallization solutions for both the apo CMD1 and the CMD1-DNA complex (**Supplementary Fig. 4f**). This might explain why CMD1 is inactive in the crystals and crystallization drops. The molecular basis for the CMD1 inactivation at the acidic conditions is unclear. It is possible that the acidic conditions might affect the protonation states of VC and/or residues involved in the binding of the metal ion, co-substrate and substrate and consequently the catalytic reaction.

We have added the results in the revised manuscript as **Supplementary Fig. 4f (Page 11)**.

Comment 3. The reader needs to see the strength of your evidence for the binding mode of VC, C and 5mC. Please show composite omit densities in the main text figures. Suppl. Fig. 4, panel B is perfect in this respect for VC, a sub-region of Suppl. Fig. 4 panels C and D would do for the bases. For the complex with 5mC dsDNA and VC, it is crucial to see the density evidence that the lactone form of VC is present in this complex, and that glyceryl transfer to 5mC has not taken place (unless it has already been shown biochemically that the reaction does not occur in crystallization conditions, see above).

Response: We thank the reviewer for the suggestion. In the revision, we have shown the composite omit density maps for the VC in the CMD1-VC complex, the flipped-out 5mC at the 3-position of the substrate strand in the CMD1-5mC-DNA complex, the flipped-out C at the 3-position of the substrate strand in the CMD1-DNA complex, and the VC and the flipped-out 5mC at the 3-position of the substrate strand in the CMD1-VC-5mC-DNA complex in the main

text Fig. 1c. The simulated annealing composite omit density maps clearly show that the bound VC is in the lactone form.

Comment 4. Like the TET reaction, the CMD1 reaction involves an oxidative decarboxylation. Comparison with the TET reaction suggests that VC is hydrolytically ring opened, and that the ring-opened form (an alpha-keto-acid) is then oxidatively decarboxylated. However, the text and Fig. 2B suggest that even in the complex with 5mC dsDNA and VC, VC is still present in the lactone form. Could this be misleading and related to hindered catalysis?

Response: We agree with the reviewer that during the catalytic reaction, the ring of VC should be hydrolytically opened, and the ring-opened form (an alpha-keto-acid) is then oxidatively decarboxylated. Our previous biochemical study showed that the CMD1 activity could not be supported by analogues or derivatives of VC (Xue et al., Nature, 2019). In addition, our further biochemical study in this work also shows that the ring-opened form of VC (2-keto-L-gulonate) and the decarboxylated intermediate (L-xylonic acid) could not support the activity of CMD1 as well (data not shown). So far, based on the available biochemical and structural data, we do not know exactly at what stage of the catalytic reaction the ring of VC is hydrolytically opened and what kind of ring-opened intermediates would be generated and employed in the catalysis, and hence we could not deduce the exact catalytic mechanism. As CMD1 is inactive in the crystals and crystallization drops (see our response to comment 2), we believe that the CMD1-5mC-DNA-VC complex represents the initial state of the catalytic reaction at which the catalysis has not taken place.

Comment 5. The Discussion on catalytic mechanism is the most interesting part of the entire work. This core part of the paper is hard to understand without a scheme that shows the reader the numbering scheme, ideally for lactone and ring-opened forms of VC. Could this be added as an inset to one of the figures? Alternatively, can you introduce the numbering scheme with the figure that shows VC?

Response: We thank the reviewer for the constructive suggestion. In the revision, we have added the numbering scheme of the carbon atoms of VC in **Fig. 2b, d**. In addition, we have added a new figure in the discussion of the catalytic mechanism and labeled the lactone and ring-opened forms of VC in the figure (**Supplementary Fig. 9**).

Comment 6. I concur with the authors that the CMD1 reaction cleaves the C1-C2 and C3-C4 bonds in vitamin C. However, I always thought that the only plausible reaction mechanism requires an attack of the 5mC methyl radical on the C4. Attack on C3 would create different reaction product. The relevant distance should therefore be the distance from the methyl group to C4, not to C3. Yet, the authors discuss both distances as if they were equally relevant for catalysis. If indeed they think so, they should explain the mechanistic rationale for why the methyl-C3 distance should matter. Otherwise, they should focus on the methyl-C4 distance.

Response: We thank the reviewer for the constructive suggestion. We fully agree with the reviewer on that the plausible mechanism of the reaction catalyzed by CMD1 requires an attack of the 5mC methyl radical on the C4 of VC, and the discussion of the distance from the methyl group to both C4 and C3 is misleading. As suggested by the reviewer, in the revision, we have

focused on the methyl-C4 distance and have made the change as follows: “These results suggest that the C1-C2 and C3-C4 bonds of VC are cleaved and the 5mC methyl radical attacks the C4 atom of VC to form the 5gmC product in the catalytic reaction. ... In the CMD1-5mC-DNA-VC complex, the C4 atom of VC is positioned from the C5-methyl group of 5mC by 5.7 Å, and the C4-C6 moiety of VC is positioned even more distantly from the C5-methyl group of 5mC.”
(Pages 23-24)

Comment 7. The methyl-C4 distance (and the geometry of radical attack) could be the highlight of the manuscript. Unfortunately, it is not clear whether this distance can be meaningfully deduced from the current structures. The authors suggest, and I completely agree, that VC decarboxylation must precede, not follow methyl radical attack on decarboxylated VC (the radical generator Fe(IV)oxo is created with participation of one oxygen atom from molecular oxygen, the other is involved in the decarboxylation). As the methyl-C4 distance is so central to the chemistry, the information on this distance should remain in the manuscript. It is important enough that the limitations of the current distance determination (possible changes of VC pose after ring opening) should be discussed (or at least mentioned).

Response: We thank the reviewer for the constructive suggestion. In the revision, we have rewritten the paragraph about the proposed catalytic mechanism and made a more extended discussion about the possible changes of the VC conformation after ring opening during the catalytic reaction as follows:

“The previous biochemical studies have shown the metabolic destiny of VC in the reaction catalyzed by CMD1²⁶. Briefly, the CO₂ is derived from the C1 atom of VC; the glyoxylic acid is derived from the C2 and C3 atoms of VC; and the glycerol moiety added to the 5mC is derived from the C4-C6 atoms of VC. These results suggest that the C1-C2 and C3-C4 bonds of VC are cleaved and the 5mC methyl radical attacks the C4 atom of VC to form the 5gmC product in the catalytic reaction. However, in the structure of CMD1-5mC-DNA-VC complex, VC is present in the lactone form and the catalytic reaction has not taken place, which is consistent with the biochemical data showing that CMD1 is inactive in the crystals and crystallization drops. Our previous biochemical studies showed that the CMD1 activity could not be supported by analogues or derivatives of VC²⁶, and our current biochemical studies also showed that the ring-opened form of VC (2-keto-L-gulonate) and the decarboxylated intermediate (L-xylonic acid) could not support the activity of CMD1 either (data not shown), indicating that the glycerol transfer reaction catalyzed by CMD1 could only take place with the lactone form of VC. Thus, the CMD1-5mC-DNA-VC complex is very likely to represent the initial state of the catalytic reaction (**Supplementary Fig. 9, step 1**). During the catalytic reaction, the lactone form of VC would open the ring at specific condition(s) and the ring-opened form of VC might maintain the mono-coordination with the metal ion or undergo some conformational changes to coordinate the metal ion in a bidentate manner similar to 2-OG (**Supplementary Fig. 9, step 2**). Then, the ring-opened form of VC is decarboxylated to generate L-xylonic acid (or other intermediate) upon the activation of O₂ by the metal ion using a similar mechanism as the other Fe²⁺/2-OG-dependent dioxygenases (**Supplementary Fig. 9, step 3**). In the CMD1-5mC-DNA-VC complex, the C4 atom of VC is positioned from the C5-methyl group of 5mC by 5.7 Å, and the C4-C6 moiety of VC is positioned even more distantly from the C5-methyl group of 5mC. Thus, it is very likely that the C4-C6 moiety of the ring-opened form or intermediate of VC would change its conformation to swing towards the C5-methyl group of

5mC (**Supplementary Fig. 9, step 4-5**), permitting the transfer of the C4-C6 moiety of VC to the C5-methyl group of 5mC. In this process, the ring-opened form or intermediate of VC might coordinate the metal ion in a bidentate mode similar to 2-OG. A modeling study shows that when *L*-xyloic acid assumes a bidentate coordination manner similar to 2-OG and is stabilized by the surrounding residues, the distance between the C4 atom of *L*-xyloic acid and the C5-methyl group of 5mC would be within 4 Å (**Supplementary Fig. 8e**), and the C4 atom could be attacked by the 5mC radical to generate 5gmC and glyoxylic acid (**Supplementary Fig. 9, step 6**). Nevertheless, based on the available biochemical and structural data, we do not know exactly at what stage of the catalytic reaction the ring of VC is hydrolytically opened and what kind of ring-opened intermediate would be generated and employed in the catalysis. Hence, the exact and detailed molecular mechanism of the catalytic reaction by CMD1 could not be deduced and warrants further biochemical and structural studies.” (**Pages 23-25**)

Comment 8. Mechanistically, the reaction CMD1 reaction requires a pyrimidine 5-methyl group. The enzyme therefore does not need to distinguish between 5mC and C to selectively transfer glyceryl to 5mC. Hence, the similar affinity of CMD1 to 5mC and C containing dsDNA is perhaps not so surprising. In contrast to C, T has a 5-methyl group placed on the pyrimidine scaffold. Hence, in rare cases the CMD1 reaction could happen with T instead of 5mC. For TETs, the Carell group has shown that similar errors indeed occur at low frequency. Does this happen with CMD1? If I interpret Fig. 2D correctly, the preference for 5mC over T is due to a favorable hydrogen bond from the N4 of 5mC to Asp350. The authors have the Asp350Asn variant (see Fig. 2C). If the wild-type protein does not do it, does this variant transfer glyceryl to T, creating a novel DNA base? While this may be an easy experiment to do given that the authors have the enzymes in hand, it is not essential experiment for the main message of the manuscript and therefore should not be required for publication.

Response: We thank the reviewer for the constructive suggestion. In the revision, we have performed enzymatic activity assays of the wild-type and D350N mutant CMD1 towards 0.5-kb 5mCpG- and CpG-containing DNAs (which contain about 25.5% T) to examine whether the wild-type CMD1 and the D350N mutant could catalyze the glyceryl transfer from VC to T. The results show that for the 5mCpG-DNA substrate, the wild-type CMD1 could catalyze the formation of 5gmC efficiently and the D350N mutant could generate a very little amount of 5gmC; and for the CpG-DNA substrate, both the wild-type and mutant CMD1 cannot catalyze the formation of 5gmC. On the other hand, in all of these assays, no new product peak could be identified at the expected position for C⁵-glyceryl-thymine and the abundance of the T peak has no notable change (**Supplementary Fig. 5e**). These results suggest that both the wild-type and the D350N mutant CMD1 cannot catalyze the reaction of a glyceryl transfer from VC to T.

These results have been added in the revised manuscript as **Supplementary Fig. 5e (Pages 14-15)**.

Technical comments. The PDB validation reports contain long lists of (moderate) steric conflicts, almost all of which are caused by implicit hydrogen atoms. At this resolution, this is not a major problem, because hydrogen positions cannot be relied on anyway. Nevertheless, the list of conflicts is oddly long. The authors may want to check their hydrogen model used in refinement, to clarify whether this is a PDB issue, or whether there are genuine conflicts that

could be avoided.

Response: We thank the reviewer for pointing out the problem. We have checked the PDB validation reports of the five structures and noticed that for the structure of the CMD1-5mC-DNA-VC complex, the all-atom clashscore in the H-added model is 8 with 110 close contacts, while for the other structures, the all-atom clashscore is between 2-4 with less than 60 close contacts. During the revision, we have re-refined the structure of the CMD1-5mC-DNA-VC complex, and the geometry of the structure model is substantially improved with the all-atom clashscore reduced to 2 and the close contacts reduced to 32 while the statistics of the structure refinement have no major changes. We have deposited the re-refined structure of the CMD1-5mC-DNA-VC complex to the PDB and the new PDB validation report is uploaded together with the revised manuscript.

Minor editorial comments:

Comment 1. Throughout: the authors have not decided between the “mC” and “5mC” nomenclature for 5-methylcytosine. Moreover, in some cases, they add the residue numbering in the structure that converts mC to mC3, according to the sequence position. If possible, this inconsistency should be avoided.

Response: We thank the reviewer for the suggestion. To avoid the inconsistency and confusion, we have unified the nomenclature for 5-methylcytosine as 5mC and removed the residue numbering of 5mC throughout the text in the revision.

Comment 2. p.3, line 48: “the epigenetics field has attracted increasing interests to the discovery of additional DNA base modifications and their biological functions.” Please reword.

Response: The sentence has been reworded as follows: “**In the past two decades, there is increasing interest in the discovery of additional DNA base modifications and their biological functions in the epigenetics field.**”

Comment 3. p. 4, line 66: “consisting of a major and a minor beta-sheets” should be “consisting of a major and a minor beta-sheet” (no plural), but please double-check with a native speaker.

Response: In the revision, we have made this change.

Comment 4. p.6, line 130: “of high AT level” should be “AT-rich”, likewise for “GC level” later on.

Response: In the revision, we have changed “high AT level” and “high GC level” to “AT-rich” and “GC-rich”, respectively.

Comment 5. p.9, paragraph: “Crystal structures of CMD1 in complexes with VC, DNA, and both VC and DNA”. Please state somewhere early in the paragraph whether the DNA contained a CpG or 5mCpG, in the substrate strand (now it takes inspection of Fig. 2 to see that a 5mC was

in the substrate strand). Fig. 3C suggests that the non-substrate strand is non-methylated. Please introduce this early on in the text.

Response: We apologize for the unclear description about the DNA substrate which might lead to the confusion of the reviewer. In the revision, we have reworded the sentence as follows: “Among these CMD1-DNA complexes, the crystals of CMD1 in complex with a 14-nt DNA which comprises a 10-bp dsDNA region and a 4-nt 3'-overhang with or without a 5mC at the 3-position on both the forward (or substrate) strand and the reverse (or non-substrate) strand diffracted X-rays to relatively higher resolution” (Page 10, line 210).

In Fig. 3c, both of the substrate strand and the non-substrate strand of the DNA are methylated and were indicated in the figure. However, the label appears to be too small to see. In the revision, we have enlarged the label of 5mC and colored it in red (Fig. 3d, old Fig. 3c) to make it clearer.

Comment 6. p. 10, line 207: “The crystals of CMD1 in complex with both DNA and VC were also obtained by soaking”. This statement is easily misunderstood. I believe that you co-crystallized CMD1 with dsDNA, and then soaked in vitamin C. Please reword.

Response: We thank the reviewer for the suggestion. The statement has been reworded as follows: “The crystals of CMD1 in complex with both DNA and VC were obtained by soaking the crystals of the CMD1-DNA complex in the reservoir solution containing VC”.

Comment 7. p. 18, line 393: “despite of the low sequence identity” should be “despite the low sequence identity”

Response: In the revision, we have made this change as suggested.

REVIEWERS' COMMENTS

Reviewer #1 (Remarks to the Author):

The authors clarified the co-substrate specificity of CMD1 towards VC but not 2OG according to the structure of the CMD1-VC complex. With more detailed enzymatic assays, the authors addressed the context preference of 5mCG for CMD1 enzyme. The structure is novel and the biochemical analyses are convincing. The structural and biochemical data lead to a model of VC-derived 5gmC DNA modification by CMD1. We support publication of this work in Nature Communications.

Reviewer #2 (Remarks to the Author):

The authors have experimentally addressed all my major concerns. It is now more convincingly shown that the metal in the active site is indeed the physiological iron and that the co-substrate is vitamin C in the lactone form. The combination of both seemed unlikely, because of the lack of a reaction in the crystal. Fortunately, the pH profile convincingly resolves the paradox, as I had hoped.

The minor issues with one of the structures (steric conflicts of implicit hydrogen atoms) have also been addressed, and the description of conclusions for the catalytic mechanism has been clarified. Some further experiments (which did not yield the results that I had hoped for) were also carried out. The study is now complete and could be published as is. I noticed only a single glitch:

-page 9, line 187: the word "iron" is duplicated

Congratulations to the authors for very nice work!